# Learning and Collusion in Multi-unit Auctions[*]

**Simina Brânzei**
Purdue University

**Mahsa Derakhshan**
Northeastern University

**Negin Golrezaei**
MIT

**Yanjun Han**
New York University

## Abstract

In a carbon auction, licenses for CO2 emissions are allocated among multiple interested players. Inspired by this setting, we consider repeated multi-unit auctions with uniform pricing, which are widely used in practice. Our contribution is to analyze these auctions in both the offline and online settings, by designing efficient bidding algorithms with low regret and giving regret lower bounds. We also analyze the quality of the equilibria in two main variants of the auction, finding that one variant is susceptible to collusion among the bidders while the other is not.

## 1 Introduction

In a multi-unit auction, a seller brings multiple identical units of a good to a group of players (buyers) interested in purchasing the items. Due to the complexity and depth of the model, the multi-unit auction has been explored in a significant volume of research [MRH89, EWK98, DN07, DLN12, FFLS12, ACP+14, DN15, BCI+05, BGN03, GML13, DHP17, BMT19] and recommended as a lens to the entire field of algorithmic mechanism design [Nis14]. The prior literature has often focused on understanding the equilibria of the auction under various solution concepts and designing revenue or welfare maximizing one-shot mechanisms.

We consider the setting where each player $i$ has a value $v_{i,j}$ for receiving a $j$-th unit in their bundle, which is reported to the auctioneer in the form of a bid $b_{i,j}$. After receiving the bids, the auctioneer computes a price $p$ per unit, then sorts the bids in descending order and allocates the $j$-th unit to the player that submitted the $j$-th highest bid, charging them $p$ for the unit. The price $p$ is set so that all units are sold and the resulting allocation has the property that each player gets their favorite bundle given their preferences.

This allocation rule represents the well-known *Walrasian mechanism* in the multi-unit setting: compute the market (aka competitive or Walrasian) equilibrium with respect to the reported valuations. The template for the mechanism is: elicit the valuations from the buyers and then assign a set of prices to the goods such that when each buyer takes their favorite bundle of goods at the given prices, the market clears, i.e. all goods are sold and there is no excessive demand for any good (see, e.g., [BLNL14]). Due to the strong fairness and efficiency properties of the market equilibrium, the resulting allocation is envy-free with respect to the reported valuations. A strand of research concentrated on achieving fair pricing in auctions, which often involves setting uniform prices for identical units [GHK+05].

Repeated auctions take place in many real world settings, such as when allocating licenses for carbon (CO2 emissions) [CK02], ads on online platforms [BBW15, GJM21], U.S. Treasury notes to investors or trade exchanges over the internet [MT15]. Repeated mechanisms often give rise to complex patterns of behavior. For example, the bidders may use the past transaction prices as a way of discovering the valuations of the competing bidders, or engage in collusion using bid rotation

---

[*]Authors are listed in alphabetical order. This work was done in part while the authors were visiting the Simons Institute for the Theory of Computing. Simina was supported in part by NSF CAREER grant CCF-2238372.

schemes since the history of bids can serve as a communication device [Aoy03]. In fact, [CCDP21] show that in markets such as Cournot oligopolies with stochastic demand, collusion is observed even when the bidders are not explicitly trying to conspire. Instead, collusion emerges naturally when the agents use AI algorithms based on Q-learning to update their strategies.

Since repeated multi-unit auctions are used in high-stakes environments such as allocating licenses for carbon [GIL20], it is paramount to understand: $(i)$ what features should the auction have in the first place?; and $(ii)$ given an auction format, how should the players bid to maximize their utility?

A good bidding strategy generally guarantees the player using it will not "regret" its strategy in hindsight, regardless of the strategies of others. There has been extensive work on understanding dynamics in auctions and games under various behavioral models of the players (see, e.g., [NCKP22, RST17, DS16, MT15, HKMN11, BR11, LB10, GJM21, GJLM21, GJL23, BF19]), but due to the complexity of understanding dynamical systems many questions remain open.

In this paper, we design efficient learning algorithms that players can use for bidding, in both the offline and online settings. We also give regret lower bounds and then analyze the quality of the equilibria in two main variants of the auction, finding that one variant is susceptible to collusion among the bidders while the other is not.

## 1.1 Model

Consider a seller with $K$ identical units of a good and a set of players $[n] = \{1, \ldots, n\}$ that have quasi-linear valuations with decreasing marginal returns. Formally, let $\mathbf{v}_i = (v_{i,1}, \ldots, v_{i,K})$ be the valuation of player $i$, where $v_{i,j} \geq 0$ is the marginal value for getting a $j$-th unit in their bundle. The value of player $i$ for getting $\ell$ units is $V_i(\ell) = \sum_{j=1}^{\ell} v_{i,j}$. Diminishing marginal returns require that $v_{i,1} \geq v_{i,2} \geq \ldots \geq v_{i,K} \ \forall i \in [n]$. A player $i$ is said to be *hungry* if $v_{i,j} > 0 \ \forall j \in [K]$.

**Auction format: uniform pricing.** The auctioneer announces $K$ units of a good for sale. Each player $i$ submits bids $\mathbf{b}_i = (b_{i,1}, \ldots, b_{i,K})$, where $b_{i,j}$ is player $i$'s bid for $j$-th unit. The auctioneer sorts the bids in decreasing order and computes a price $p$. Then for each $j = 1, \ldots, K$, the auctioneer allocates the $j$-th unit to the player that submitted the $j$-th highest bid, charging them $p$ for the unit. If multiple bids are equal to each other, ties are broken in lexicographic order of the player names. [2]

We consider two variants of the auction:

    (i) *the $K$-th price auction*, where the price per unit is set to the $K$-th highest bid;

    (ii) *the $(K + 1)$-st price auction*, where the price is set to the $(K + 1)$-st highest bid.

Both of these variants implement the market equilibrium with respect to the reported valuations of the players, which are represented by their bid vectors. The $K$-th price format selects the maximum possible market equilibrium price, namely the highest possible price per unit at which all the goods are sold and each player purchases their favorite bundle given their utility. The $(K + 1)$-st format selects the minimum possible market equilibrium price.

*Allocation, Price, Utility.* An allocation $\mathbf{z}$ is an assignment of units to the players such that $z_i \geq 0$ is the number of units received by player $i$ and $\sum_{j=1}^{n} z_j = K$.

Given bid profile $\mathbf{b}$, let $p(\mathbf{b})$ be the *price* per unit and $\mathbf{x}(\mathbf{b}) = (x_1(\mathbf{b}), \ldots, x_n(\mathbf{b}))$ the *allocation* obtained when the auction is run with bids $\mathbf{b}$, where $x_i(\mathbf{b}) \in \{0, \ldots, K\}$ is the number of units received by player $i$. The *utility* of player $i$ at $\mathbf{b}$ is: $u_i(\mathbf{b}) = V_i(x_i(\mathbf{b})) - p(\mathbf{b}) \cdot x_i(\mathbf{b})$.

*Revenue and Welfare.* The revenue obtained by the auctioneer at a bid profile $\mathbf{b}$ equals $K \cdot p(\mathbf{b})$. The social welfare of the bidders at $\mathbf{b}$ is $SW(\mathbf{b}) = \sum_{i=1}^{n} V_i(x_i(\mathbf{b}))$.

*Notation.* W.l.o.g., we have $b_{i,1} \geq \ldots \geq b_{i,K}$ for each bid profile $\mathbf{b}$ is such that and $i \in [n]$. Given a bid profile $\mathbf{b} = (\mathbf{b}_1, \ldots, \mathbf{b}_n)$, let $\mathbf{b}_{-i}$ denote the bid profile of everyone except player $i$. The bid profile where player $i$ uses a bid vector $\boldsymbol{\beta}$ and the other players bid $\mathbf{b}_{-i}$ is denoted $(\boldsymbol{\beta}, \mathbf{b}_{-i})$.

An illustration of how the auction works is given in Example 1.

---

[2]In other words, after the players submit their bids, the auctioneer arranges these bids in descending order $(c_1, \ldots, c_j, \ldots, c_{n \cdot K})$ and then allocates unit 1 to the bidder with bid $c_1$, unit 2 to the bidder with bid $c_2$, and so on.

**Example 1.** *Let $K = 3$ and $n = 2$. Suppose the valuations are $\mathbf{v}_1 = (5, 2)$ and $\mathbf{v}_2 = (4, 1)$. If the players submit bids $\mathbf{b}_1 = (2, 1)$ and $\mathbf{b}_2 = (3, 2)$, the bid are sorted in the order $(b_{2,1}, b_{1,1}, b_{2,2}, b_{1,2})$. Then the allocation is $x_1 = 1$ and $x_2 = 2$.*

- *Under the $K$-th price auction, the price is set to $p = b_{2,2} = 2$. The utilities of the players are $u_1(\mathbf{b}) = V_1(1) - p = 5 - 2 = 3$ and $u_2(\mathbf{b}) = V_2(2) - 2 \cdot p = (4 + 1) - 2 \cdot 2 = 1$.*

- *Under the $(K + 1)$-st price auction, the price is set to $p = b_{1,2} = 1$. The utilities of the players are $u_1(\mathbf{b}) = V_1(1) - p = 5 - 1 = 4$ and $u_2(\mathbf{b}) = V_2(2) - 2 \cdot p = (4+1) - 2 \cdot 1 = 3$.*

**Repeated setting.** We consider a repeated setting, where the auction is run multiple times among the same set of players, who can adjust their bids based on the outcomes from previous rounds.

Formally, in each round $t = 1, 2, \ldots, T$, the next steps take place:

- The auctioneer announces $K$ units for sale. Each player $i \in [n]$ submits bid vector $\mathbf{b}_i^t = (b_{i,1}^t, \ldots, b_{i,K}^t)$ privately to the auctioneer, where $b_{i,j}^t$ is player $i$'s bid for a $j$-th unit at time $t$. Then the auctioneer runs the auction with bids $\mathbf{b}^t = (\mathbf{b}_1^t, \ldots, \mathbf{b}_n^t)$ and reveals information (feedback) about the outcome to the players. We consider two feedback models for the information revealed at the end of round $t$:
    - *Full information:* All the bids $\mathbf{b}^t$ are public knowledge.
    - *Bandit feedback:* The price $p(\mathbf{b}^t)$ is the only public knowledge; additionally, each player $i$ privately learns their own allocation $x_i(\mathbf{b}^t)$.

Thus the allocation at time $t$ solely depends on the bids submitted by players in round $t$, excluding any prior rounds. [3]

## 1.2 Our Results

The next sections summarize our results for both the $K$-th and $(K + 1)$-st price variants.

### 1.2.1 Offline Setting

In the offline setting, we are given a player $i$ with valuation $\mathbf{v}_i$ and a history $H_{-i} = (\mathbf{b}_{-i}^1, \ldots, \mathbf{b}_{-i}^T)$ containing the bids submitted by the other players in past auctions. Here we restrict the bidding space to a discrete domain $\mathbb{D} = \{k \cdot \varepsilon \mid k \in \mathbb{N}\}$, for some $\varepsilon > 0$.

The goal is to find a fixed bid vector $\mathbf{b}_i^* = (b_{i,1}^*, \ldots, b_{i,K}^*)$ that maximizes player $i$'s cumulative utility on the data set given by the history $H_{-i}$, that is, $\mathbf{b}_i^* = \arg\max_{\boldsymbol{\beta} \in \mathbb{D}^K} \sum_{t=1}^T u_i(\boldsymbol{\beta}, \mathbf{b}_{-i}^t)$.

The offline problem is challenging because the decision (bid) space is exponentially large and hence naively experimenting with every possible bid vector leads to impractical algorithms.

We overcome this challenge by relying on the structural properties of the offline problem. At a high level, we carefully design a weighted directed acyclic graph (DAG) and identify a bijective map between paths from source to sink in the graph and bid profiles of player $i$. Since a maximum weight path in a DAG can be found in polynomial time, this yields a polynomial time algorithm for computing an optimum bid vector.

**Theorem 1** (informal)**.** *Computing an optimum bid vector for one player in the offline setting is equivalent to finding a maximum-weight path in a DAG and can be solved in polynomial time.*

Theorem 1 applies to both versions of the auction, with $K$-th and $(K + 1)$-st highest price.

### 1.2.2 Online Setting

In the online setting, the players run learning algorithms to update their bids as they participate in the auction over time. The first scenario we consider is that of full information feedback, where at the end of each round $t$ the auctioneer makes public all the bids $\mathbf{b}^t$ submitted in that round.

---

[3]For example, consider $K = 2$ units and $n = 2$ players. Suppose that in round $t = 5$, player 1 submits the bid vector $b_1^5 = (4, 2)$ and player 2 submits $b_2^5 = (5, 3)$. The auctioneer sorts these bids in decreasing order, obtaining the vector $(5, 4, 3, 2)$. Then the allocation in round 5 is to give the first unit to player 1 and the second unit to player 2; the allocation disregards any events from preceding rounds.

Building on the offline analysis, we design an efficient online learning algorithm for bidding, which guarantees low regret for each player using it. The main idea is to run multiplicative weight updates, where the experts are paths in the DAG from the offline setting. Each such path corresponds to a bid profile for the learner. A challenge is that the number of paths in the graph (and so experts) is exponential. Nevertheless, we can achieve a polynomial runtime by representing the experts implicitly, using a type of weight pushing algorithm based on the method of path kernels [TW03].

**Theorem 2** (Full information feedback, upper bound). *For each player $i$ and time horizon $T$, there is an algorithm for bidding in the repeated auction with full information feedback that runs in time $O(T^2)$ and guarantees the player's regret is bounded by $O(v_{i,1} \cdot \sqrt{TK^3 \log T})$.*

To the best of our knowledge, our paper is the first to connect the path kernel setting with auctions and obtain efficient algorithms for learning in auctions via this connection.

We also consider bandit feedback, which limits the amount of information the players learn about each other: at the end of each round the auctioneer announces publicly only the resulting price, and then privately tells each player their allocation. Bandit feedback could be relevant for reducing the amount of collusion among the players in repeated auction environments [HP89].

**Theorem 3** (Bandit feedback, upper bound). *For each player $i$ and time horizon $T$, there is an algorithm for bidding in the repeated auction with bandit feedback that runs in $O(KT + K^{-5/4}T^{7/4})$ and guarantees the player's regret is bounded by $O(v_{i,1} \cdot \min\{\sqrt[4]{T^3 K^7 \log T}, KT\})$.*

Although our bidding policy is similar to the EXP3-type algorithm in [GLLO07], and our computational efficiency relies on the path kernel methods in [TW03], a major difference is that our edge weights are *heterogeneous*, i.e. the weights of different edges could be of different scales. To circumvent this issue, we need to use a different estimator for the weight under bandit feedback which in turn also changes the technical analysis.

We complement the upper bounds with the following regret lower bound, where the construction of the hard instance relies heavily on the specific structure of our auction format.

**Theorem 4** (Lower bound, full information and bandit feedback). *Let $K \geq 2$. Suppose the auction is run for $T$ rounds. Then for each strategy of player $i$, under both full information and bandit feedback, there exists a bid sequence $\{\mathbf{b}^t_{-i}\}_{t=1}^T$ by the other players such that the expected regret of player $i$ is at least $c \cdot v_{i,1} K \sqrt{T}$, where $c > 0$ is an absolute constant.*

Theorems 2, 3, and 4 apply to both variants of the auction (with $K$-th and $(K+1)$-st highest price).

### 1.2.3 Equilibrium Analysis

We also analyze the quality of the Nash equilibria reached in the worst case in the static version of the auction, as they naturally apply to the empirical distribution of joint actions when the players use sub-linear regret learning algorithms.

The $(K+1)$-st price auction has been observed to have low or even zero revenue equilibria [Wil79, KN04, ACP$^+$14, BW20]. With $n > K$ hungry players, the next phenomena occur:

- *in the $(K+1)$-st price auction:* every allocation $\mathbf{z}$ that allocates all the units can be implemented in a pure Nash equilibrium $\mathbf{b}$ with price $\varepsilon$, for every small enough $\varepsilon \geq 0$.
- *in the $K$-th price auction:* no pure Nash equilibrium with arbitrarily low price exists.

Our contribution is to show that the zero-price equilibria of the $(K+1)$-st highest price auction are very stable. We do this by considering the core solution concept, which models how groups of players can coordinate.

In our setting, a strategy profile is *core-stable* if no group $S$ of players can deviate by simultaneously changing their strategies such that each player in $S$ weakly improves their utility *and* the improvement is strict for at least one player in $S$. The players outside $S$ are assumed to have neutral reactions to the deviation, that is, they keep their strategy profiles unchanged. This is consistent with the Nash equilibrium solution concept, where only the deviating player changes their strategy.

A body of literature studied the core in auctions when the auctioneer can collude together with the bidders (see, e.g., [DM08]). However, as is standard in mechanism design, it is also meaningful to

consider the scenario where the auctioneer sets the auction format and then the bidders can strategize and potentially collude among themselves, without the auctioneer; this setting is our focus.

First, we consider the core without transfers, where a strategy profile is simply a bid profile $\mathbf{b}$.

**Theorem 5** (Core without transfers). *Consider $K$ units and $n > K$ hungry players. The core without transfers of the $(K + 1)$-st auction can be characterized as follows:*

- *every bid profile $\mathbf{b}$ that is core-stable has price zero (i.e. $p(\mathbf{b}) = 0$);*

- *for every allocation $\mathbf{z}$, there is a core-stable bid profile $\mathbf{b}$ with price zero that implements $\mathbf{z}$ (i.e. $\mathbf{x}(\mathbf{b}) = \mathbf{z}$ and $p(\mathbf{b}) = 0$).*

If a bid profile $\mathbf{b}$ is core-stable, then it is also a Nash equilibrium, for the simple reason that if no group of players can deviate simultaneously from $\mathbf{b}$, then the group consisting of one player cannot find an improving deviation either. Thus Theorem 5 says that the equilibria with price zero are the only ones that remain stable when also allowing deviations by groups of players. Moreover, the theorem also says there are many inefficient core-stable equilibria, all with price zero.

Second, we also consider the core with transfers, where the players can make monetary transfers to each other. A strategy profile in this setting is given by a pair $(\mathbf{b}, \mathbf{t})$, where $\mathbf{b}$ is a bid profile and $\mathbf{t}$ is a profile of monetary transfers, such that $t_{i,j} \geq 0$ is the monetary transfer (payment) of player $i$ to player $j$ (e.g., player $i$ could pay player $j$ so that $j$ keeps its bids low).

**Theorem 6** (Core with transfers). *Consider $K$ units and $n > K$ hungry players. The core with transfers of the $(K + 1)$-st auction can be characterized as follows.*

- *Let $(\mathbf{b}, \mathbf{t})$ be an arbitrary tuple of bids and transfers that is core stable. Then the allocation $\mathbf{x}(\mathbf{b})$ maximizes social welfare, the price is zero (i.e. $p(\mathbf{b}) = 0$), and there are no transfers between the players (i.e. $\mathbf{t} = 0$).*

To see why Theorem 6 holds, if the price is positive at some strategy profile, the players with "highest values" can pay the other players to lower their bids. When the price reaches zero, they can start withdrawing the payments. Thus the only strategy profiles in the core with transfers are those where the players with "highest values" win and pay zero. The difference from Theorem 5 is that transfers allow the players with highest values to force getting their desired items at price zero, which they cannot enforce without transfers. Nevertheless, even without transfers, the price remains zero.

***Remarks on collusion.*** Given that the zero price equilibria of the $(K + 1)$-st price auction are very stable and easy to find, one can expect they can be determined by the bidders, for example in settings with few bidders who know each other. These equilibria can be easily implemented in repeated settings once the bidders agree on their strategies.

Strikingly, these zero price equilibria have in fact been observed empirically in repeated auctions for fishing quotas in the Faroe Islands, where even inexperienced bidders were able to collude and enforce them; see details in [LMT18]. In the carbon setting, the Northeast US auction is based on the $(K + 1)$-st price format [GIL20] and has very low prices as well.

In contrast, the $K$-th price auction format does not have any Nash equilibria with price zero, let alone core-stable ones. Thus the $K$-th price auction format may be preferable to the $(K + 1)$-st price auction in settings with high stakes, such as selling rights for carbon emissions. Another remedy, also discussed in [LMT18], is to make the number of units a random variable.

## 2 Related Work

Our work is related to the rich literature on combinatorial auctions (e.g., [DVV03, BN07, CSS07]) and more specifically auctions for identical goods, which are widely used in practice in various settings: Treasury Auctions [GI05, BS00], Procurement Auctions [AC05], Wholesale Electricity Markets [TSM08, FR03, FvdFH06], Carbon Auctions [GS13, SS17, GIL20]. The two most prevalent multi-unit auctions are uniform price auctions and pay-as-bid auctions that are widely studied in the literature either empirically (e.g., [BZ93, MA98, BB01, HG13]) or theoretically under the classical Bayesian setting (e.g., [AHP13, GIL20, BILL23]). In particular, there have been several studies that aim to compare uniform price auctions versus pay-as-bid auctions (e.g.,

[KCPT01, FR03, SBLS04, LMMM11, ACP+14]) in terms of revenue and welfare under Nash equilibria, and the possibility of collusion. Our work contributes to this literature by studying uniform price auctions the angle of designing bidding algorithms for the players.

Designing learning algorithms in auctions has attracted significant attention in recent years. Several papers study the problem of designing learning/bandit algorithms to optimize reserve prices in auctions over time (e.g., [KN14, MR16, MM16, CD17, DHL+17, GJL23, GJM21, GJLM21]). There are papers that have focused on designing bandit algorithms to optimize bidding strategies in single-unit second price and first price auctions (e.g., [WPR16, BMSW18, FPS18, HZF+20, HZW20, BGM+22]). Prior to our work, such problem was not studied for multi-unit uniform-price auctions.

Leveraging structural information in designing bandit algorithms has been the topic of the rich literature, which includes linear reward structures [DHK08, RT10, MRT09, LS17], Lipschitz reward structures [MCP14, MLS18], convex reward distribution structure [VPG20], and combinatorial structures [AR08, UNK10, CBL12, ABL14, CWY13, CTPL15, ZLW19, KV05, NGW+21]. Our work is closest to the works that aim to exploit combinatorial structures. Within this literature, several papers leverage DP-based or MDP-based structures (e.g., [TW03, GLLO07, ZN13, RM21]).

The multi-unit auction setting was studied in a large body of literature on auctions [DLN12, BCI+05, DN07, DL14, DN15], where the focus has been on designing truthful auctions with good approximations to some desired objective, such as the social welfare or the revenue.

Uniform pricing represents a type of fair pricing, since when everyone gets charged the same price per unit, no bidder envies another bidder's allocation. Fairness is an important property that is at odds with maximizing revenue, since for the purpose of improving revenue the auctioneer is better off charging higher prices to the buyers that show more interest in the goods. However, there is evidence that buyers are unhappy with discriminatory prices (see, e.g., [AS08]), which led to a body of literature focused on designing fair auction mechanisms [GHK+05, FGL14, FFLS12, CAEFF16, Sek17, AVGN17, BFMZ17, FMTV20, DGJ+22]. Collusion in auctions was also studied both experimentally and theoretically in various models (e.g., [GN92, vdFH93, GNR15]).

## 3 Offline Setting

In the offline setting, we are given a player $i$ with valuation $\mathbf{v}_i$ and a history $H_{-i} = (\mathbf{b}_{-i}^1, \ldots, \mathbf{b}_{-i}^T)$ with the bids of other players in rounds 1 through $T$. We assume the bids are restricted to a discrete domain $\mathbb{D} = \{k \cdot \varepsilon \mid k \in \mathbb{N}\}$, for some $\varepsilon > 0$. The goal is to find an optimum fixed bid vector in hindsight: $\mathbf{b}_i^* = \arg\max_{\boldsymbol{\beta} \in \mathbb{D}^K} \sum_{t=1}^{T} u_i(\boldsymbol{\beta}, \mathbf{b}_{-i}^t)$, where we recall that $u_i(\boldsymbol{\beta}, \mathbf{b}_{-i}^t)$ is the utility of player $i$ at the bid profile $(\boldsymbol{\beta}, \mathbf{b}_{-i}^t)$. We will see the maximum in the definition of $\mathbf{b}_i^*$ is well defined since there is an optimum bid vector $\boldsymbol{\beta}$ with entries at most $\varepsilon$ above the maximum historical bid.

For any $\varepsilon > 0$, the optimal solution on the discrete domain $\mathbb{D}$ also yields an $(\varepsilon \cdot K)$-optimal strategy on the continuous domain. The offline problem we study here has some resemblance to the problems studied in [RW16, DGPL19, DPS21]. There, the goal is to optimize reserve prices in VCG auctions while having access to a dataset of historical bids.[4]

To design a polynomial time algorithm for solving the offline problem, we define a set $\mathcal{S}_i$ of "candidate" bids for player $i$, which has polynomial size in the history, player $i$'s valuation, and $1/\varepsilon$:

$$\mathcal{S}_i = \{0\} \cup \left\{ b_{j,k}^t \mid j \in [n] \setminus \{i\}, k \in [K] \right\} \cup \left\{ b_{j,k}^t + \varepsilon \mid j \in [n] \setminus \{i\}, k \in [K] \right\}. \quad (1)$$

**Observation 1.** *Player $i$ has an optimum bid vector $\boldsymbol{\beta} = (\beta_1, \ldots, \beta_K) \in \mathbb{D}^K$ with $\beta_j \in \mathcal{S}_i$ for all $j \in [K]$, where $\mathcal{S}_i$ is defined in equation* (1).

The proof of Observation 1 is simple and deferred to Appendix A. Next we construct a directed acyclic graph (DAG) $G_i$ that will be used to compute an optimum bid vector for player $i$.

**Definition 1** (The graph $G_i$). *Given valuation $\mathbf{v}_i$ of a player $i$, history $H_{-i} = (\mathbf{b}_{-i}^1, \ldots, \mathbf{b}_{-i}^T)$, $\varepsilon > 0$, and set $\mathcal{S}_i$ from* (1), *define a graph $G_i$ with the following features.*

---

[4][RW16] also presents an online version of their offline algorithms under full information feedback; see also [NGW+21] for an online version of [RW16]'s algorithm for both full information and bandit settings using Blackwell approachability [Bla56].

**Vertices.** *Create a vertex $z_{s,j}$ for each $s \in \mathcal{S}_i$ and $j \in [K]$. We say vertex $z_{s,j}$ is in layer $j$. Add source $z_-$ and sink $z_+$.*

**Edges.** *For each index $j \in [K-1]$ and pair of bids $r, s \in \mathcal{S}_i$ with $r \geq s$, create a directed edge from vertex $z_{r,j}$ to vertex $z_{s,j+1}$. Moreover, add edges from source $z_-$ to each node in layer 1 and from each node in layer $K$ to the sink $z_+$.*

**Edge weights.** *For each edge $e = (z_{r,j}, z_{s,j+1})$ or $e = (z_{r,K}, z_+)$, let $\boldsymbol{\beta} = (\beta_1, \ldots, \beta_K) \in \mathcal{S}_i^K$ be a bid vector with $\beta_j = r$ and $\beta_{j+1} = s$ (we define $s = 0$ if $j = K$). For each $t \in [T]$, let $\mathbf{h}^t = (\boldsymbol{\beta}, \mathbf{b}_{-i}^t)$. Define the weight of edge $e$ as*

$$w_e = \sum_{t=1}^{T} \mathbb{1}_{\{x_i(\mathbf{h}^t) \geq j\}} (v_{i,j} - r) + j \Big[ \mathbb{1}_{\{x_i(\mathbf{h}^t) > j\}} (r - s) + \mathbb{1}_{\{x_i(\mathbf{h}^t) = j\}} \big(r - p(\mathbf{h}^t)\big) \Big], \quad (2)$$

*recalling that $p(\mathbf{h}^t)$ and $\mathbf{x}(\mathbf{h}^t)$ are the price and allocation at $\mathbf{h}^t$, respectively [5]. The edges incoming from $z_-$ have weight zero.*

A pictorial illustration of the graph $G_i$ is displayed in Figure 1.

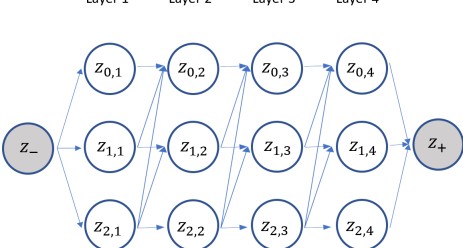

Figure 1: Example of DAG as in Definition 1. Here, we have $K = 4$ units and the set of candidate bids for player $i$ is $\mathcal{S}_i = \{0, 1, 2\}$. The graph has a source $z_-$, a sink $z_+$, and nodes $z_{r,j}$ for all $r \in \mathcal{S}_i$ and $j \in [4]$. Edges are of the form $(z_{r,j}, z_{s,j+1})$ for all $j < 4$ and $r, s \in \mathcal{S}_i$ with $r \geq s$. There are also edges from the source $z_-$ to nodes $z_{j,1}$ and from nodes $z_{j,4}$ to the sink $z_+$ $\forall j \in \mathcal{S}_i$.

At a first glance, it may seem as though the weight $w_e$ of an edge $e$ depends on the whole bid vector $\boldsymbol{\beta}$ since we use the value of $x_i(\mathbf{h}^t)$ in its definition. However, this is not the case. Meaning that, this formula gives us the same value for any choice of $\boldsymbol{\beta}$ that satisfies $\beta_j = r$ and $\beta_{j+1} = s$. This is due to the fact that the values of variables $\mathbb{1}_{\{x_i(\mathbf{h}^t) \geq j\}}$, $\mathbb{1}_{\{x_i(\mathbf{h}^t) > j\}}$, and $\mathbb{1}_{\{x_i(\mathbf{h}^t) = j\}}$ can be determined only knowing the values of $\beta_{j+1}$ and $\beta_j$ (without knowing the rest of the vector $\boldsymbol{\beta}$).

We will show that computing an optimum strategy for the player is equivalent to finding a maximum-weight path in the DAG $G_i$ from Definition 1.

**Theorem 1** (formal version). *Suppose we are given a number $n$ of players, number $K$ of units, valuation $\mathbf{v}_i$ of player $i$, discretization level $\varepsilon > 0$, and bid history $H_{-i} = (\mathbf{b}_{-i}^1, \ldots, \mathbf{b}_{-i}^T)$ by players other than $i$. Then an optimum bid vector for player $i$ can be computed in polynomial time in the input parameters.*

The proof of Theorem 1 is in Appendix A, together with the other omitted proofs of this section.

At a high level, for any bid vector $\boldsymbol{\beta}$, we first write the cumulative utility of player $i$ when playing $\boldsymbol{\beta}$ while the others play according to $\mathbf{b}_{-i}^t$. Then observe the cummulative utility can be rewritten as a sum of $K$ terms, so that each term only depends on $j$, $\beta_j$, and $\beta_{j+1}$. Create a layered DAG as in Definition 1, where the weight of edge $(\beta_j, j)$ to $(\beta_{j+1}, j+1)$ is the $j$-th term of the sum above, with special handling for edges involving the source or the sink. The entire graph can be computed in polynomial time.

Then the proof shows there is a bijection between the set of bid vectors of the player and the set of paths from the source to the sink in the graph. Thus the solution to the offline problem is obtained by computing a maximum weight path in the graph and returning the corresponding bid vector.

The algorithm for the offline problem is summarized in the next figure.

---

[5]That is, $x_k(\mathbf{h}^t)$ is the number of units received by player $k$ at allocation $\mathbf{x}(\mathbf{h}^t)$.

**Input:** Player $i$ with valuation $\mathbf{v}_i = (v_{i,1}, \ldots, v_{i,K})$, a history $H_{-i} = (\mathbf{b}_{-i}^1, \ldots, \mathbf{b}_{-i}^T)$ of bids submitted by players other than $i$, and a grid $\mathbb{D} = \{k \cdot \varepsilon \mid k \in \mathbb{N}\}$ with allowed bid values.

**Output:** $\mathbf{b}_i^* = \arg\max_{\boldsymbol{\beta} \in \mathbb{D}^K} \sum_{t=1}^T u_i(\boldsymbol{\beta}, \mathbf{b}_{-i}^t)$.

- Compute set $\mathcal{S}_i$ as defined in equation (1). By Observation 1, player $i$ has an optimum bid vector $\mathbf{b}_i^*$ with $b_{i,j}^* \in \mathcal{S}_i$ for each $j \in [K]$.
- Construct directed acyclic graph $G_i$ as in Definition 1. By Theorem 1, there is a bijective map between paths from the source to the sink in $G_i$ and bid vectors $\boldsymbol{\beta} \in \mathcal{S}_i$ of player $i$.
- Compute a maximum-weight path in graph $G_i$ and output the corresponding bid vector.

algorithm 1: Optimum Strategy for the Offline Problem.

# 4  Online Setting

In the online setting, we consider the repeated auction where in each round $t$ the auctioneer announces $K$ units for sale. Then the players privately submit the bids $\mathbf{b}^t$ and the auctioneer computes the outcome. At the end of round $t$, the auctioneer gives *full information* or *bandit* feedback. The information available to player $i$ at the beginning of each round $t$ is the history:
$H_i^{t-1} = (\mathbf{b}_i^1, \cdots, \mathbf{b}_i^{t-1}) \cup H_{-i}^{t-1}$, where [6]

$$
H_{-i}^{t-1} = \begin{cases} \left(\mathbf{b}_{-i}^1, \cdots, \mathbf{b}_{-i}^{t-1}\right) & \text{in the full information setting,} \\ \left(p(\mathbf{b}^1), \cdots, p(\mathbf{b}^{t-1})\right) & \text{in the bandit setting.} \end{cases}
$$

A pure *strategy* for player $i$ at time $t$ is a function $\pi_i^t : H_i^{t-1} \to \mathbb{R}^K$, where $H_i^{t-1}$ is the history available to player $i$ at the beginning of round $t$. Thus the pure strategy tells player $i$ what bid vector to submit next given the information observed by the player about the previous rounds. A mixed strategy is a probability distribution over the set of pure strategies. Consequently, the bid vector $\mathbf{b}_i^t$ submitted by player $i$ at time $t$ is the realization of the mixed strategy $\pi_i^t(H_i^{t-1})$. We denote by $\pi_i = (\pi_i^1, \cdots, \pi_i^T)$ the overall strategy of player $i$ over the entire time horizon.

**Regret.**    From now on we fix an arbitrary player $i$. Given a bidding strategy $\pi_i$ of player $i$, the regret of the player is defined with respect to a history $H_{-i}^T$ of bids by other players:

$$
\mathrm{Reg}_i(\pi_i, H_{-i}^T) = \max_{\boldsymbol{\beta} \in \mathcal{S}_i^K} \sum_{t=1}^T \sum_{j=1}^K \left(v_{i,j} - p(\boldsymbol{\beta}, \mathbf{b}_{-i}^t)\right) \cdot \mathbb{1}_{\{x_i(\boldsymbol{\beta}, \mathbf{b}_{-i}^t) \geq j\}}
$$
$$
- \sum_{t=1}^T \mathbb{E}_{\mathbf{b}_i^t \sim \pi_i^t(H_i^{t-1})} \left[ \sum_{j=1}^K \left(v_{i,j} - p(\mathbf{b}^t)\right) \cdot \mathbb{1}_{\{x_i(\mathbf{b}^t) \geq j\}} \right] . \quad (3)
$$

For the purpose of equipping player $i$ with a bidding algorithm, we will think of the other players as adversarial, and aim to achieve small regret regardless of $H_{-i}^T$. A way to interpret the regret is that player $i$ is competing against an oracle that (a) has the perfect knowledge of the history of bids, but (b) is restricted to submit a time-invariant bid $\boldsymbol{\beta}$. The regret quantifies how much worse player $i$ does compared to the oracle.

Before studying the feedback models in greater detail, we replace the set $\mathcal{S}_i$ of candidate bids for player $i$ from equation (1) of the offline section by a coarser set $\mathcal{S}_\varepsilon = \{\varepsilon, 2\varepsilon, \cdots, \lceil v_{i,1}/\varepsilon \rceil \varepsilon\}$.

## 4.1  Full information

In the full information scenario, at the end of each round $t$ the auctioneer announces the bids $\mathbf{b}^t$ submitted by the players in round $t$. We define a graph $G^t$ that forms the basis of the learning

---

[6]We omit the allocation information in the history, for it is not informative when other players are adversarial: from the price information, player $i$ knows her own allocation; treating all other players as a giant adversarial player, that player gets the rest of the allocation.

algorithm for the player. The graph $G^t$ is the same as in the offline setting (Definition 1), except for the next differences:

- The vertex $z_{s,j}$ is indexed by $s \in \mathcal{S}_\varepsilon$, instead of $\mathcal{S}_i$;

- The edge weights are based only on the current round $t$, rather than the sum over all rounds in the offline setting. More specifically, for edge $e = (z_{r,j}, z_{s,j+1})$ or $e = (z_{r,K}, z_+)$, set

$$w^t(e) = \mathbb{1}_{\{x_i(\mathbf{h}^t) \geq j\}} (v_{i,j} - r) + j\Big[ \mathbb{1}_{\{x_i(\mathbf{h}^t) > j\}} (r - s) + \mathbb{1}_{\{x_i(\mathbf{h}^t) = j\}} \big(r - p(\mathbf{h}^t)\big) \Big]. \quad (4)$$

The formal definition is deferred to Definition 2 in Appendix B. This DAG structure enables us to formulate the learning problem as an online maximum weight path problem, and therefore treat each path as an expert and apply online learning algorithms to compete with the best expert. Theorem 2 shows such an algorithm (Algorithm 2) works for both $K$-th and $(K+1)$-st price auctions.

---

**Input:** time horizon $T$, valuation $\mathbf{v}_i = (v_{i,1}, \cdots, v_{i,K})$ of the player $i$
**Output:** bid vectors $(\mathbf{b}_i^1, \cdots, \mathbf{b}_i^T)$.
Set the resolution parameter $\varepsilon = v_{i,1}\sqrt{K/T}$, and learning rate $\eta = \sqrt{\log T}/(v_{i,1}\sqrt{KT})$.
For every $t = 1, \cdots, T$:

- Construct a graph $G^t$ as in Definition 2, initially without weights on the edges.
- If $t = 1$: for each edge $e$, initialize an edge probability as

$$\phi^1(e) = \begin{cases} 1/|\mathcal{S}_\varepsilon| & \text{if } e = (z_-, z_{r,1}); \\ 1/|\{r' \in \mathcal{S}_\varepsilon \mid r' \leq r\}| & \text{if } e = (z_{r,j}, z_{s,j+1}), r \geq s; \\ 1 & \text{if } e = (z_{r,K}, z_+). \end{cases}$$

- If $t \geq 2$:
  - starting from $\Gamma^{t-1}(z_+) = 1$, recursively compute in the bottom-up order

  $$\Gamma^{t-1}(u) = \sum_{v:e=(u,v) \in E} \phi^{t-1}(e) \cdot \exp(\eta \cdot w^{t-1}(e)) \cdot \Gamma^{t-1}(v), \quad \forall u \in V(G^t);$$

  - for each edge $e$, update the edge probability

  $$\phi^t(e) = \phi^{t-1}(e) \cdot \exp\big(\eta \cdot w^{t-1}(e)\big) \cdot \frac{\Gamma^{t-1}(v)}{\Gamma^{t-1}(u)}, \quad \text{for all } e = (u,v) \in E(G^t). \quad (5)$$

- For $j = 1, 2, \cdots, K$: sample $b_{i,j}^t \sim \phi^t(z_{b_{i,j-1}^t, j-1}, \cdot)$, where $z_{b_{i,0}^t, 0} := z_-$.
- The player observes $\mathbf{b}_{-i}^t$, and sets the edge weights $w^t(e)$ in $G^t$ according to (4).

---

algorithm 2: Selecting bids using the weight pushing algorithm in [TW03].

In Algorithm 2, inspired by the path kernel algorithm in [TW03], we define $\phi^t(u, \cdot)$ as a probability distribution over the outneighbors of vertex $u$. To determine the bid vector in each round, we use a random walk starting from the source $z_-$, where the learner selects a random neighbor $z_1$ of $z_-$ using $\phi^t(z_-, \cdot)$, then selects a random neighbor $z_2$ of $z_1$ using $\phi^t(z_1, \cdot)$, and so on, until reaching the sink $z_+$. Remarkably, as we prove in Theorem 2, the distribution of the resulting path $\mathfrak{p}^t$ is the same as that in the Multiplicative Weight Update (Hedge) algorithm [LW94].

Thus while the number of paths in the DAG (corresponding to experts in the Hedge algorithm) is exponentially large, we can run the Hedge algorithm with polynomial space and time complexity by leveraging the DAG structure. We do not need to keep track of the weights of every path, but only need to store and update the weights of every *edge* in the DAG using the variables $\phi^t(e)$ and the recursive formula in equation (5). To accomplish this, we use the variables $\Gamma^t(u)$, for every vertex $u$ in the DAG, which can be viewed as the weights of paths starting at vertex $u$.

We obtain the next theorem , the proof of which can be found in Appendix B.

**Theorem 2.** *For each player $i$ and time horizon $T$, under full-information feedback, Algorithm 2 runs in time $O(T^2)$ and guarantees the player's regret is at most $O(v_{i,1}\sqrt{TK^3 \log T})$.*

## 4.2 Bandit feedback

The next theorem presents an upper bound on the regret under the bandit feedback. Similar to the full-information case, the algorithm we design works for both the $K$-th and $(K+1)$-st price auctions.

**Theorem 3.** *For each player $i$ and time horizon $T$, there is an algorithm for bidding in the repeated auction with bandit feedback that runs in $O(KT + K^{-5/4}T^{7/4})$ and guarantees the player's regret is bounded by $O(v_{i,1} \cdot \min\{\sqrt[4]{T^3 K^7 \log T}, KT\})$.*

We describe the algorithm below, while the proof of the theorem can be found in Appendix B.

The bidding strategy and its efficient implementation are the same as those in the proof of Theorem 2, with the only difference as follows. Under bandit feedback, we are not able to compute $w^t(e)$ for every edge $e$ at the end of time $t$; instead, we are able to compute $w^t(e)$ for every edge on the chosen path $\mathfrak{p}^t$ at time $t$. This observation motivates us to use the next unbiased estimator $\widehat{w}^t(e)$:

$$\widehat{w}^t(e) = \overline{w}(e) - \frac{\overline{w}(e) - w^t(e)}{p^t(e)} \mathbb{1}_{\{e \in \mathfrak{p}^t\}}, \qquad \text{with } p^t(e) = \sum_{\mathfrak{p}:e \in \mathfrak{p}} P^t(\mathfrak{p}), \tag{6}$$

where $P^t$ is the player's action distribution over paths, and for each edge $e$:

$$\overline{w}(e) = \begin{cases} v_{i,1} - r + j(r-s) & \text{if } e = (z_{r,j}, z_{s,j+1}); \\ v_{i,1} - r + Kr & \text{if } e = (z_{r,K}, z_+); \\ 0 & \text{if } e = (z_-, z_{r,1}). \end{cases}$$

The resulting algorithm is exactly the same as Algorithm 2, with the only difference being that all $w^t$ are replaced by $\widehat{w}^t$, with an additional step (6) for the computation of $\widehat{w}^t$.

The regret analysis will appear similar to that in [GLLO07], but a key difference is that we no longer have $w^t(e) \le v_{i,1}$ for every edge $e$ in our setting. Instead, we apply an edge-dependent upper bound $w^t(e) \le \overline{w}(e)$, and use the property that $\sum_{e \in \mathfrak{p}} \overline{w}(e) \le K v_{i,1}$ for every path $\mathfrak{p}$ from source to sink.

## 4.3 Regret lower bounds

The following theorem establishes an $\Omega(K\sqrt{T})$ lower bound on the minimax regret of a single bidder in the full information scenario. Clearly the same lower bound also holds for bandit feedback.

**Theorem 4** (restated, formal). *Let $K \ge 2$. For any policy $\pi_i$ used by player $i$, there exists a bid sequence $\{\mathbf{b}^t_{-i}\}_{t=1}^T$ for the other players such that the expected regret in (3) satisfies $\mathbb{E}[\mathrm{Reg}_i(\pi_i, \{\mathbf{b}^t_{-i}\}_{t=1}^T)] \ge c v_{i,1} K\sqrt{T}$, where $c > 0$ is an absolute constant.*

The proof of Theorem 4 is in Appendix B. The idea is to employ the celebrated Le Cam's two-point method, but the construction of the hard instance is rather delicate to reflect the regret dependence on $K$. We also remark that this regret lower bound still exhibits a gap compared with the current upper bound in Theorem 2, while it seems hard to extend the current two-point construction to Fano-type lower bound arguments. It is an outstanding question to close this gap.

## 5 Concluding Remarks

Several open questions arise from this work. For example, is the regret $\Theta(K\sqrt{T})$ in both the full information and bandit setting? In the full information setting, the challenge in obtaining an algorithm with regret $O(K\sqrt{T})$ is that the expert distribution is not a product distribution over the layers, but rather only close to a product. Furthermore, how does offline and online learning take place in repeated auctions when the units are not identical, the valuations do not necessarily exhibit diminishing returns, or the pricing rule is different (e.g. in the first price setting)?

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

## Roadmap to the appendix

Appendix A includes the proofs omitted from Section 3 (Offline Setting). Appendix B includes the proofs omitted from Section 4 (Online Setting). Appendix C includes the equilibrium analysis. A few results from prior work that we invoke are in Appendix D.

## A   Appendix: Offline Setting

In this section we include the proofs omitted from the main text for the offline setting.

**Observation 1 (restated).**   *Player $i$ has an optimum bid vector $\boldsymbol{\beta} = (\beta_1, \ldots, \beta_K) \in \mathbb{D}^K$ with $\beta_j \in \mathcal{S}_i$ for all $j \in [K]$.*

*Proof.* Let $\mathbf{c} = (c_1, \ldots, c_K)$ be an arbitrary optimum strategy for player $i$. Suppose $\mathbf{c} \notin \mathcal{S}_i^K$.

We use $\mathbf{c}$ to construct an optimum bid vector $\boldsymbol{\beta} \in \mathcal{S}_i^K$ as follows. For each $j \in [K]$:

- If $c_j \in \mathcal{S}_i$, then set $\beta_j = c_j$.

- Else, set $\beta_j = \max\{y \in \mathcal{S}_i \mid y \leq c_j\}$.

Then in each round $t$, player $i$ gets the same allocation when playing $\boldsymbol{\beta}$ as it does when playing $\mathbf{c}$ and the others play $\mathbf{b}_{-i}^t$ since the ordering of the owners of the bids is the same under $(\mathbf{c}, \mathbf{b}_{-i}^t)$ as it is under $(\boldsymbol{\beta}, \mathbf{b}_{-i}^t)$; moreover, the price weakly decreases in each round $t$. Thus player $i$'s utility weakly improves. Since $\mathbf{c}$ was an optimal strategy for player $i$, it follows that $\boldsymbol{\beta}$ is also an optimal strategy for player $i$ and, moreover, $\boldsymbol{\beta} \in \mathcal{S}_i^K$ as required.   $\square$

Next we show how to find an optimal bid vector in polynomial time. The proof uses several lemmas, which are proved after the theorem.

**Theorem 1 (restated, formal).** *Suppose we are given a number $n$ of players, number $K$ of units, valuation $\mathbf{v}_i$ of player $i$, discretization level $\varepsilon > 0$, and bid history $H_{-i} = (\mathbf{b}_{-i}^1, \ldots, \mathbf{b}_{-i}^T)$ by players other than $i$. Then an optimum bid vector for player $i$ can be computed in polynomial time in the input parameters.*

*Proof of Theorem 1.* Compute the set $\mathcal{S}_i$ given by equation (1) and the graph $G_i$ from Definition 1. The proof has several steps as follows.

$G_i$ **is a DAG.**   All the edges in $G_i$ flow from the source $z_-$ to the nodes from layer 1, then from the nodes in layer $j$ to those in layer $j+1$ (i.e. of the form $(z_{r,j}, z_{s,j+1})$ for all $j \in [K-1]$ and $r, s \in \mathcal{S}_i$ with $r \geq s$), and finally from all the nodes in layer $K$ to the sink $z_+$. A cycle would require at least one back edge, but such edges do not exist.

**Bijective map between bid vectors and paths from source to sink in $G_i$.**   To each bid vector $\boldsymbol{\beta} = (\beta_1, \ldots, \beta_K) \in \mathcal{S}_i^K$, associate the following path in $G_i$: $P(\boldsymbol{\beta}) = \left(z_-, z_{\beta_1,1}, \ldots, z_{\beta_K,K}, z_+\right)$.

We show next the map $P$ is a bijection from the set $\mathcal{S}_i$ of candidate bid vectors for player $i$ to the set of paths from source to sink in $G_i$. Consider arbitrary bid vector $\boldsymbol{\beta} = (\beta_1, \ldots, \beta_K) \in \mathcal{S}_i^K$. By definition of a bid vector, we have $\beta_1 \geq \ldots \geq \beta_K$. Then $P(\boldsymbol{\beta}) = (z_-, z_{\beta_1,1}, \ldots, z_{\beta_K,K}, z_+)$ is a valid path in $G_i$.

Conversely, since $G_i$ has an edge $(z_{r,j}, z_{s,j+1})$ if and only if $r \geq s$, each path from source to sink in $G_i$ has the form $Q = (z_-, z_{\beta_1,1}, \ldots, z_{\beta_K,K}, z_+)$ for some numbers $\beta_1, \ldots, \beta_K \in \mathcal{S}_i$, and so it can be mapped to bid vector $\boldsymbol{\beta} = (\beta_1, \ldots, \beta_K)$. Thus $Q = P(\boldsymbol{\beta})$, and so $P$ is a bijective map.

**Utility of player $i$ and weight of a path of length $K$ in $G_i$.**   Let $\boldsymbol{\beta} = (\beta_1, \ldots, \beta_K) \in \mathcal{S}_i^K$. Additionally, let $\beta_{K+1} = 0$. The total utility of player $i$ when bidding $\boldsymbol{\beta}$ in each round $t$ while the others bid $\mathbf{b}_{-i}^t$ is $U_i(\boldsymbol{\beta}) = \sum_{t=1}^T u_i(\mathbf{h}^t)$, where $\mathbf{h}^t = (\boldsymbol{\beta}, \mathbf{b}_{-i}^t) \; \forall t \in [T]$.

Let $P(\boldsymbol{\beta}) = (z_-, z_{\beta_1,1}, \ldots, z_{\beta_K,K}, z_+)$ be the path in $G_i$ corresponding to $\boldsymbol{\beta}$. The edges of the path $P(\boldsymbol{\beta})$ are $(z_-, z_{\beta_1,1}), (z_{\beta_1,1}, z_{\beta_2,2}), \ldots, (z_{\beta_{K-1},K-1}, z_{\beta_K,K}), (z_{\beta_K,K}, z_+)$, while the weight of the path is equal to the sum of its edges. Summing equation (2), which gives the weight of an edge, across all edges of $P(\boldsymbol{\beta})$ implies that the weight of path $P(\boldsymbol{\beta})$ is equal to

$$w(P(\boldsymbol{\beta})) = \sum_{j=1}^{K} \sum_{t=1}^{T} \left[ \mathbb{1}_{\{x_i(\mathbf{h}^t) \geq j\}} (v_{i,j} - \beta_j) + j \left( \mathbb{1}_{\{x_i(\mathbf{h}^t) > j\}} (\beta_j - \beta_{j+1}) + \mathbb{1}_{\{x_i(\mathbf{h}^t) = j\}} (\beta_j - p(\mathbf{h}^t)) \right) \right].$$
(7)

We claim that $U_i(\boldsymbol{\beta}) = w(P(\boldsymbol{\beta}))$. The high level idea is to rewrite the utility to "spread it" across the edges of the path corresponding to bid profile $\boldsymbol{\beta}$. Towards this end, recall the utility of player $i$ at strategy profile $\mathbf{h}^t = (\boldsymbol{\beta}, \mathbf{b}^t_{-i})$ is

$$u_i(\mathbf{h}^t) = \sum_{j=1}^{K} \mathbb{1}_{\{x_i(\mathbf{h}^t) \geq j\}} (v_{i,j} - p(\mathbf{h}^t)).$$
(8)

By Lemma 1, equation (8) is equivalent to

$$u_i(\mathbf{h}^t) = \sum_{j=1}^{K} \left[ \mathbb{1}_{\{x_i(\mathbf{h}^t) \geq j\}} (v_{i,j} - \beta_j) + j \left( \mathbb{1}_{\{x_i(\mathbf{h}^t) > j\}} (\beta_j - \beta_{j+1}) + \mathbb{1}_{\{x_i(\mathbf{h}^t) = j\}} (\beta_j - p(\mathbf{h}^t)) \right) \right].$$
(9)

Summing $u_i(\mathbf{h}^t)$ over all rounds $t$ gives

$$\begin{aligned}
U_i(\boldsymbol{\beta}) &= \sum_{t=1}^{T} u_i(\mathbf{h}^t) \\
&= \sum_{t=1}^{T} \sum_{j=1}^{K} \left[ \mathbb{1}_{\{x_i(\mathbf{h}^t) \geq j\}} (v_{i,j} - \beta_j) + j \left( \mathbb{1}_{\{x_i(\mathbf{h}^t) > j\}} (\beta_j - \beta_{j+1}) + \mathbb{1}_{\{x_i(\mathbf{h}^t) = j\}} (\beta_j - p(\mathbf{h}^t)) \right) \right] \\
&\qquad\qquad\qquad\qquad\qquad\qquad\qquad\qquad\qquad\qquad\qquad\qquad\qquad\qquad \text{(By equation (9))} \\
&= \sum_{j=1}^{K} \sum_{t=1}^{T} \left[ \mathbb{1}_{\{x_i(\mathbf{h}^t) \geq j\}} (v_{i,j} - \beta_j) + j \left( \mathbb{1}_{\{x_i(\mathbf{h}^t) > j\}} (\beta_j - \beta_{j+1}) + \mathbb{1}_{\{x_i(\mathbf{h}^t) = j\}} (\beta_j - p(\mathbf{h}^t)) \right) \right] \\
&\qquad\qquad\qquad\qquad\qquad\qquad\qquad\qquad\qquad\qquad\qquad\qquad \text{(Swapping the order of summation)} \\
&= w(P(\boldsymbol{\beta})). \qquad\qquad\qquad\qquad\qquad\qquad\qquad\qquad\qquad\qquad\qquad\qquad \text{(By equation (7))}
\end{aligned}$$

Thus the weight of the path $P(\boldsymbol{\beta})$ is equal to the utility of player $i$ from bidding $\boldsymbol{\beta}$. The implication is that to find an optimum bid vector, it suffices to compute a maximum weight path in $G_i$.

**Computing a maximum weight path in $G_i$.** The graph $G_i$ is a DAG with a number of vertices of $G_i$ that is polynomial in the input parameters. By Lemma 2, the edge weights of $G_i$ can be computed in polynomial time.

A maximum weight path in a DAG can be computed in polynomial time by changing every weight to its negation to obtain a graph $-G$. Since $G$ has no cycles, the graph $-G$ has no negative cycles. Thus running a shortest path algorithm on $-G$ will yield a longest (i.e. maximum weight) path on $G$ in polynomial time, as required.

The unique bid vector corresponding to the maximum weight path found can then be recovered in time $O(K)$ and it represents an optimum bid vector for player $i$. $\qquad\square$

**Lemma 1.** *In the setting of Theorem 1, for each bid profile $\boldsymbol{\beta} \in \mathcal{S}_i^K$ and round $t \in [T]$, define $\mathbf{h}^t = (\boldsymbol{\beta}, \mathbf{b}^t_{-i})$. Then we have*

$$u_i(\mathbf{h}^t) = \sum_{j=1}^{K} \left[ \mathbb{1}_{\{x_i(\mathbf{h}^t) \geq j\}} (v_{i,j} - \beta_j) + j \left( \mathbb{1}_{\{x_i(\mathbf{h}^t) > j\}} (\beta_j - \beta_{j+1}) + \mathbb{1}_{\{x_i(\mathbf{h}^t) = j\}} (\beta_j - p(\mathbf{h}^t)) \right) \right].$$
(10)

*Proof.* Given bid profile $\boldsymbol{\beta} = (\beta_1, \ldots, \beta_K)$, we also define $\beta_{K+1} = 0$.

If $x_i(\mathbf{h}^t) = 0$ then both sides of equation (10) are zero, so the statement holds. Thus it remains to prove equation (10) when $x_i(\mathbf{h}^t) > 0$. Let $u_{i,j}(\mathbf{h}^t) = \mathbb{1}_{\{x_i(\mathbf{h}^t) \geq j\}}(v_{i,j} - p(\mathbf{h}^t))$ for each $j \in [K]$. The term $u_{i,j}(\mathbf{h}^t)$ represents the amount of utility obtained from the $j$-th unit acquired by player $i$ at price $p(\mathbf{h}^t)$, so $u_i(\mathbf{h}^t) = \sum_{j=1}^{K} u_{i,j}(\mathbf{h}^t)$.

We first show that for each $j \in [K]$:

$$u_{i,j}(\mathbf{h}^t) = \mathbb{1}_{\{x_i(\mathbf{h}^t) \geq j\}}(v_{i,j} - \beta_j) + \sum_{k=j}^{K}\left[\mathbb{1}_{\{x_i(\mathbf{h}^t) > k\}}(\beta_k - \beta_{k+1}) + \mathbb{1}_{\{x_i(\mathbf{h}^t) = k\}}(\beta_k - p(\mathbf{h}^t))\right].$$

(11)

To prove (11), we will rewrite $u_{i,j}(\mathbf{h}^t)$ by considering three cases and writing a unified expression for all of them.

1. $x_i(\mathbf{h}^t) = j$. Then

$$\begin{aligned}
u_{i,j}(\mathbf{h}^t) &= v_{i,j} - p(\mathbf{h}^t) \\
&= (v_{i,j} - \beta_j) + (\beta_j - p(\mathbf{h}^t)) \\
&= \mathbb{1}_{\{x_i(\mathbf{h}^t) \geq j\}}(v_{i,j} - \beta_j) + \mathbb{1}_{\{x_i(\mathbf{h}^t) = j\}}(\beta_j - p(\mathbf{h}^t) \\
&= \mathbb{1}_{\{x_i(\mathbf{h}^t) \geq j\}}(v_{i,j} - \beta_j) + \sum_{k=j}^{K}\left[\mathbb{1}_{\{x_i(\mathbf{h}^t) > k\}}(\beta_k - \beta_{k+1}) + \mathbb{1}_{\{x_i(\mathbf{h}^t) = k\}}(\beta_k - p(\mathbf{h}^t))\right].
\end{aligned}$$

(Since $\mathbb{1}_{\{x_i(\mathbf{h}^t) > k\}} = 0 \; \forall k \geq j$ and $\mathbb{1}_{\{x_i(\mathbf{h}^t) = k\}} = 1$ if and only if $k = j$.)

2. $x_i(\mathbf{h}^t) > j$. Then $x_i(\mathbf{h}^t) = j + \ell$, for some $\ell \in \{1, \ldots, K - j\}$. We have

$$\begin{aligned}
u_{i,j}(\mathbf{h}^t) &= v_{i,j} - p(\mathbf{h}^t) \\
&= (v_{i,j} - \beta_j) + (\beta_j - \beta_{j+\ell}) + (\beta_{j+\ell} - p(\mathbf{h}^t)) \\
&= (v_{i,j} - \beta_j) + \left(\sum_{k=j}^{j+\ell-1} \beta_k - \beta_{k+1}\right) + (\beta_{j+\ell} - p(\mathbf{h}^t)).
\end{aligned}$$

(12)

We are in the case where $\mathbb{1}_{\{x_i(\mathbf{h}^t) \geq j\}} = 1$, $\mathbb{1}_{\{x_i(\mathbf{h}^t) > k\}} = 1$ if and only if $k \in \{j, \ldots, j + \ell - 1\}$, and $\mathbb{1}_{\{x_i(\mathbf{h}^t) = k\}} = 1$ if and only if $k = j + \ell$. Adding indicators to the terms in (12) gives

$$u_{i,j}(\mathbf{h}^t) = \mathbb{1}_{\{x_i(\mathbf{h}^t) \geq j\}}(v_{i,j} - \beta_j) + \sum_{k=j}^{K}\left[\mathbb{1}_{\{x_i(\mathbf{h}^t) > k\}}(\beta_k - \beta_{k+1}) + \mathbb{1}_{\{x_i(\mathbf{h}^t) = k\}}(\beta_k - p(\mathbf{h}^t))\right].$$

3. $x_i(\mathbf{h}^t) < j$. Then $\mathbb{1}_{\{x_i(\mathbf{h}^t) \geq j\}} = 0$, $\mathbb{1}_{\{x_i(\mathbf{h}^t) > k\}} = 0$ for all $k \in \{j, \ldots, K\}$, and $\mathbb{1}_{\{x_i(\mathbf{h}^t) = k\}} = 0$ for all $k \in \{j, \ldots, K\}$. Thus we trivially have the identity required by the lemma statement since $u_{i,j}(\mathbf{h}^t) = 0$ and the right hand side of (11) is zero as well.

Thus equation (11) holds in all three cases as required.

For each $i \in [n]$ and $j \in [K]$, define

$$A_{i,j} = \sum_{k=j}^{K}\left[\mathbb{1}_{\{x_i(\mathbf{h}^t) > k\}}(\beta_k - \beta_{k+1}) + \mathbb{1}_{\{x_i(\mathbf{h}^t) = k\}}(\beta_k - p(\mathbf{h}^t))\right].$$

(13)

Then equation (11) is equivalent to

$$u_{i,j}(\mathbf{h}^t) = \mathbb{1}_{\{x_i(\mathbf{h}^t) \geq j\}}(v_{i,j} - \beta_j) + A_{i,j},$$

(14)

Summing equation (13) over all $j \in [K]$ gives

$$\sum_{j=1}^{K} A_{i,j} = \sum_{j=1}^{K} \sum_{k=j}^{K} \left[ \mathbb{1}_{\{x_i(\mathbf{h}^t)>k\}}(\beta_k - \beta_{k+1}) + \mathbb{1}_{\{x_i(\mathbf{h}^t)=k\}}(\beta_k - p(\mathbf{h}^t)) \right] \tag{15}$$

$$\overset{a}{=} \sum_{k=1}^{K} \sum_{j=1}^{k} \left[ \mathbb{1}_{\{x_i(\mathbf{h}^t)>k\}}(\beta_k - \beta_{k+1}) + \mathbb{1}_{\{x_i(\mathbf{h}^t)=k\}}(\beta_k - p(\mathbf{h}^t)) \right] \tag{16}$$

$$= \sum_{k=1}^{K} (\beta_k - \beta_{k+1}) \sum_{j=1}^{k} \mathbb{1}_{\{x_i(\mathbf{h}^t)>k\}} + \sum_{k=1}^{K} (\beta_k - p(\mathbf{h}^t)) \sum_{j=1}^{k} \mathbb{1}_{\{x_i(\mathbf{h}^t)=k\}} \tag{17}$$

$$= \sum_{k=1}^{K} (\beta_k - \beta_{k+1}) \cdot k \cdot \mathbb{1}_{\{x_i(\mathbf{h}^t)>k\}} + \sum_{k=1}^{K} (\beta_k - p(\mathbf{h}^t)) \cdot k \cdot \mathbb{1}_{\{x_i(\mathbf{h}^t)=k\}} \tag{18}$$

where equation (a) holds because we change the order of the double summations and in the double summations, we consider any (integer) pair of $(j,k)$ such that $1 \leq k \leq j \leq K$.

Then we can rewrite the utility $u_i(\mathbf{h}^t)$ as

$$u_i(\mathbf{h}^t) = \sum_{j=1}^{K} u_{i,j}(\mathbf{h}^t)$$

$$= \sum_{j=1}^{K} \left[ \mathbb{1}_{\{x_i(\mathbf{h}^t)\geq j\}}(v_{i,j} - \beta_j) + A_{i,j} \right] \hspace{2cm} \text{(By equation (11))}$$

$$= \sum_{j=1}^{K} \mathbb{1}_{\{x_i(\mathbf{h}^t)\geq j\}}(v_{i,j} - \beta_j) + \sum_{j=1}^{K}(\beta_j - \beta_{j+1}) \cdot j \cdot \mathbb{1}_{\{x_i(\mathbf{h}^t)>j\}} + \sum_{k=1}^{K}(\beta_j - p(\mathbf{h}^t)) \cdot j \cdot \mathbb{1}_{\{x_i(\mathbf{h}^t)=j\}}$$

$$\hspace{10cm} \text{(By equation (18))}$$

$$= \sum_{j=1}^{K} \left[ \mathbb{1}_{\{x_i(\mathbf{h}^t)\geq j\}}(v_{i,j} - \beta_j) + j\left( \mathbb{1}_{\{x_i(\mathbf{h}^t)>j\}}(\beta_j - \beta_{j+1}) + \mathbb{1}_{\{x_i(\mathbf{h}^t)=j\}}(\beta_j - p(\mathbf{h}^t)) \right) \right],$$

as required by the lemma statement. $\qquad \square$

**Lemma 2.** *In the setting of Theorem 1, the edge weights of the graph $G_i$ can be computed in polynomial time.*

*Proof.* Recall the graph $G_i$ is constructed given as parameters the number $n$ of players, the number of units $K$, a player $i$ with valuation $\mathbf{v}_i$, discretization level $\varepsilon > 0$, and a bid history $H_{-i} = (\mathbf{b}_{-i}^1, \ldots, \mathbf{b}_{-i}^T)$ by players other than $i$. Thus the goal is to show the edge weights of $G_i$ can be computed in polynomial time in the bit length of these parameters.

Towards this end, we will show there exist efficiently computable values $I_{>j}^t, I_{\geq j}^t, I_j^t \in \{0,1\}$ and $q^t \in \mathcal{S}_i$ such that the weight $w_e$ of edge $e = (z_{r,j}, z_{s,j+1})$ is equal to

$$w_e = \sum_{t=1}^{T} I_{\geq j}^t \left( v_{i,j} - r \right) + j\left( I_{>j}^t \left( r - s \right) + I_j^t \left( r - q^t \right) \right). \tag{19}$$

Roughly, $q^t$ will correspond to the price and $I_{>j}^t, I_{\geq j}^t, I_j^t$ will tell whether player $i$ gets more than $j$ units, at least $j$ units, or exactly $j$ units, respectively, at some profile $\boldsymbol{\beta} = (\beta_1, \ldots, \beta_K) \in \mathcal{S}_i$ with $\beta_j = r$ and $\beta_{j+1} = s$. The choice of $\boldsymbol{\beta}$ in will not matter, as long as $\beta_j = r$ and $\beta_{j+1} = s$.

We prove equation (19) in several steps:

***Step (i).*** For each $t \in [T]$, define $\Gamma^t : \mathbb{R} \to \mathbb{R}$, where

$$\Gamma^t(x) = \left| \left\{ (\ell, j) \mid \left( b_{\ell,j}^t > x \text{ and } \ell \in [n] \setminus \{i\}, j \in [K] \right) \right. \right.$$

$$\left. \left. \text{or } \left( b_{\ell,j}^t = x \text{ and } \ell \in [n], \ell < i, j \in [K] \right) \right\} \right|.$$

Thus $\Gamma^t(x)$ counts the bids in profile $\mathbf{b}^t_{-i}$ that would have priority to a bid of value $x$ submitted by player $i$ (i.e. it counts bids strictly higher than $x$ and submitted by players other than $i$, as well as bids equal to $x$ but submitted by players lexicographically before $i$).

***Step (ii).*** Recall the edge is denoted $e = (z_{r,j}, z_{s,j+1})$. Let $\boldsymbol{\beta} \in \mathcal{S}_i$ be an arbitrary bid profile with $\beta_j = r$ and $\beta_j = s$. Also define $\beta_{K+1} = 0$.

If $\Gamma^t(s) < K - j$, then at $\mathbf{h}^t = (\boldsymbol{\beta}, \mathbf{b}^t_{-i})$ player $i$ receives one unit for each of the bids $\beta_1, \ldots, \beta_{j+1}$, since there are at most $K - j - 1$ bids of other players that have higher priority than player $i$'s highest $j + 1$ bids. Else, player $i$ does not get more than $j$ units. Thus $I^t_{>j} = \mathbb{1}_{\{x_i(\mathbf{h}^t) > j\}}$.

***Step (iii).*** If $\Gamma^t(r) \leq K - j$, then player $i$ receives at least $j$ units at $(\boldsymbol{\beta}, \mathbf{b}^t_{-i})$. The corresponding indicator is

$$I^t_{\geq j} = \mathbb{1}_{\{\Gamma^t(s) \leq K-j\}} = \mathbb{1}_{\{x_i(\mathbf{h}^t) \geq j\}}.$$

***Step (iv).*** If $\Gamma^t(r) \leq K - j$ and $\Gamma^t(s) > K - j$, then player $i$ receives exactly $j$ units at $(\boldsymbol{\beta}, \mathbf{b}^t_{-i})$. The indicator is

$$I^t_j = \mathbb{1}_{\{\Gamma^t(r) \leq K-j\}} \cdot \mathbb{1}_{\{\Gamma^t(s) > K-j\}} = \mathbb{1}_{\{x_i(\mathbf{h}^t) = j\}}.$$

Now suppose $I^t_j = 1$. Then we show the price $q^t = p(\mathbf{h}^t)$ can be computed precisely without knowing the whole bid $\boldsymbol{\beta}$.

Consider the multiset of bids $B = \mathbf{b}^t_{-i} \cup \{\beta_1, \ldots, \beta_{j+1}\}$, recalling $\beta_{K+1}$ was defined as zero. Sort $B$ in descending order. We know the top $j - 1$ bids of player $i$ are winning, so the price is not determined by any of them. Thus the price is determined by $\beta_j = r$, $\beta_{j+1} = s$, or by one of the bids in $\mathbf{b}^t_{-i}$. Remove elements $\beta_1, \ldots, \beta_{j-1}$ from $B$ and set the price as follows:

- ***Case (iv.a).*** For $(K + 1)$-st price auction: set $q^t$ to the $(K + 2 - j)^{\text{th}}$ highest value in $B$.

- ***Case (iv.b).*** For $K$-th price auction: set $q^t$ to the $(K + 1 - j)^{\text{th}}$ highest value in $B$.

***Step (v).*** Combining steps (i-iv), the weight of edge $e = (z_{r,j}, z_{s,j+1})$ can be rewritten as

$$w_e = \sum_{t=1}^T \mathbb{1}_{\{x_i(\mathbf{h}^t) \geq j\}} (v_{i,j} - r) + j \left[ \mathbb{1}_{\{x_i(\mathbf{h}^t) > j\}} (r - s) + \mathbb{1}_{\{x_i(\mathbf{h}^t) = j\}} (r - p(\mathbf{h}^t)) \right]$$

(By Definition 1)

$$= \sum_{t=1}^T I^t_{\geq j} (v_{i,j} - r) + j \left[ I^t_{>j} (r - s) + I^t_j (r - q^t) \right].$$

(By steps (i-iv))

Thus the weight of each edge can be computed in polynomial time as required. □

# B   Appendix: Online Setting

In this section we include the material omitted from the main text for the online setting.

Before studying the full information and bandit feedback models in greater detail, we replace the set $\mathcal{S}_i$ of candidate bids for player $i$ from equation (1) of the offline section by a coarser set

$$\mathcal{S}_\varepsilon = \{\varepsilon, 2\varepsilon, \cdots, \lceil v_{i,1}/\varepsilon \rceil \varepsilon\}.$$

The regret, which was defined in equation (3), depends on whether the model is the $K$-th or $(K+1)$-st price auction; however the next lemma holds for both variants of the auction.

**Lemma 3.** *For all $\varepsilon > 0$, let $\mathrm{Reg}_i(\pi_i, H^T_{-i}, \varepsilon)$ be the counterpart of (3) where $\mathcal{S}_i$ is replaced by the set $\mathcal{S}_\varepsilon$. Then $\mathrm{Reg}_i(\pi_i, H^T_{-i}) \leq \mathrm{Reg}_i(\pi_i, H^T_{-i}, \varepsilon) + TK\varepsilon$.*

*Proof.* First observe that it is not beneficial to bid above $v_{i,1}$, so we can assume without loss of generality that the maximizer $\boldsymbol{\beta}$ in (3) satisfies $\beta_j \leq v_{i,1}$ for all $j \in [K]$. Now we convert $\boldsymbol{\beta}$ into another bid vector $\boldsymbol{\beta}^\varepsilon \in \mathcal{S}_\varepsilon^K$ as follows: for each $j \in [K]$, let

$$\beta_j^\varepsilon = \min\{y \in \mathcal{S}_\varepsilon \mid y \geq \beta_j\}.$$

Clearly $\beta_j \leq \beta_j^\varepsilon \leq \beta_j + \varepsilon$.

Then we claim that the next inequalities hold:

$$p(\boldsymbol{\beta}^\varepsilon, \mathbf{b}^t_{-i}) \leq p(\boldsymbol{\beta}, \mathbf{b}^t_{-i}) + \varepsilon; \tag{20}$$

$$\mathbb{1}_{\{x_i(\boldsymbol{\beta}^\varepsilon, \mathbf{b}^t_{-i}) \geq j\}} \geq \mathbb{1}_{\{x_i(\boldsymbol{\beta}, \mathbf{b}^t_{-i}) \geq j\}}. \tag{21}$$

Inequality (20) follows since

- $p(\mathbf{b})$ is either the $K$-th or the $(K+1)$-st largest element of $\mathbf{b}$, and

- increasing each entry of $\mathbf{b}$ by at most $\varepsilon$ can only increase $p(\mathbf{b})$ by no more than $\varepsilon$.

Inequality (21) is due to the fact that bidding a higher price can only help to win the unit.

Consequently, we have

$$\sum_{t=1}^{T}\sum_{j=1}^{K} \left(v_{i,j} - p(\boldsymbol{\beta}, \mathbf{b}^t_{-i})\right) \cdot \mathbb{1}_{\{x_i(\boldsymbol{\beta}, \mathbf{b}^t_{-i}) \geq j\}} \leq \sum_{t=1}^{T}\sum_{j=1}^{K} \left(v_{i,j} - p(\boldsymbol{\beta}, \mathbf{b}^t_{-i})\right) \cdot \mathbb{1}_{\{x_i(\boldsymbol{\beta}^\varepsilon, \mathbf{b}^t_{-i}) \geq j\}}$$

$$\leq \sum_{t=1}^{T}\sum_{j=1}^{K} \left(v_{i,j} - p(\boldsymbol{\beta}^\varepsilon, \mathbf{b}^t_{-i}) - \varepsilon\right) \cdot \mathbb{1}_{\{x_i(\boldsymbol{\beta}^\varepsilon, \mathbf{b}^t_{-i}) \geq j\}}$$

$$\leq \sum_{t=1}^{T}\sum_{j=1}^{K} \left(v_{i,j} - p(\boldsymbol{\beta}^\varepsilon, \mathbf{b}^t_{-i})\right) \cdot \mathbb{1}_{\{x_i(\boldsymbol{\beta}^\varepsilon, \mathbf{b}^t_{-i}) \geq j\}} + TK\varepsilon.$$

This gives the desired statement of the lemma. $\qquad\square$

## B.1   Full Information Feedback

In this section we include the omitted details for the full information setting.

We begin with the formal definition of the graph $G^t = (V, E, w^t)$ used by the online learning algorithm with full information feedback.

**Definition 2** (The graph $G^t$). *Given valuation $\mathbf{v}_i$ of player $i$, bid profile $\mathbf{b}^t_{-i}$ of the players at round $t$, and $\varepsilon > 0$, construct a graph $G^t = (V, E, w^t)$ as follows.*

- ***Vertices.** Create a vertex $z_{s,j}$ for each $s \in \mathcal{S}_\varepsilon$ and index $j \in [K]$. We say vertex $z_{s,j}$ is in layer $j$. Add source $z_-$ and sink $z_+$.*

- ***Edges.** For each index $j \in [K-1]$ and pair of bids $r, s \in \mathcal{S}_\varepsilon$ with $r \geq s$, create a directed edge from vertex $z_{r,j}$ to vertex $z_{s,j+1}$. Moreover, add edges from source $z_-$ to each node in layer 1 and from each node in layer $K$ to the sink $z_+$.*

- **Edge weights.** *For each edge $e = (z_{r,j}, z_{s,j+1})$ or $e = (z_{r,K}, z_+)$, let $\boldsymbol{\beta} = (\beta_1, \ldots, \beta_K) \in \mathcal{S}_\varepsilon^K$ be a bid vector with $\beta_j = r$ and $\beta_{j+1} = s$ (we define $s = 0$ if $j = K$). Define the weight of edge $e$ as*

$$w^t(e) = \mathbb{1}_{\{x_i(\boldsymbol{\beta}, \mathbf{b}_{-i}^t) \geq j\}} (v_{i,j} - r) + j \Big[ \mathbb{1}_{\{x_i(\boldsymbol{\beta}, \mathbf{b}_{-i}^t) > j\}} (r - s) + \mathbb{1}_{\{x_i(\mathbf{h}^t) = j\}} \big( r - p(\boldsymbol{\beta}, \mathbf{b}_{-i}^t) \big) \Big].$$

*The edges incoming from $z_-$ have weight zero.*

The next observation follows immediately from the definition.

**Observation 2.** *The next properties hold:*

- *The weight $w^t(e)$ of each edge $e$ of $G^t$ can be computed efficiently (see Lemma 2).*

- *There is a bijective map between bid vectors $\boldsymbol{\beta} \in \mathcal{S}_\varepsilon^K$ of player $i$ and paths from the source to the sink of $G^t$ (see proof of Theorem 1).*

Next we include the main theorem with its proof for the online learning algorithm under full information feedback.

**Theorem 2 (restated).** *For each player $i$ and time horizon $T$, under full-information feedback, Algorithm 2 runs in time $O(T^2)$ and guarantees the player's regret is at most $O\left(v_{i,1}\sqrt{TK^3 \log T}\right)$.*

*Proof.* The detailed algorithm description is presented as Algorithm 2. At a high level, the proof has three parts: $(i)$ showing that Algorithm 2 is a correct implementation of the Hedge algorithm where each expert is a path from source to sink in the DAG $G^t$ for time $t$, $(ii)$ bounding its regret for an appropriate choice of the learning rate, and $(iii)$ bounding the runtime.

**Step I: Algorithm 2 is a correct implementation of Hedge.** We show that Algorithm 2 is the same as the Hedge algorithm in which every path in the DAG is equivalent to an expert in the Hedge algorithm.

To do so, for each vertex $u$ in the DAG, let $\phi^t(u, \cdot)$ be a probability distribution over the outneighbors of $u$. Then, given the recursive sampling of the bids based on the probability distribution $\phi^t$'s, the probability that a path $\mathfrak{p}$ is chosen in Algorithm 2 is equal to

$$P^t(\mathfrak{p}) = \prod_{e \in \mathfrak{p}} \phi^t(e), \tag{22}$$

where $\phi^t$'s are updated in equation (5), which is restated below:

$$\phi^t(e) = \phi^{t-1}(e) \cdot \exp\left(\eta \cdot w^{t-1}(e)\right) \cdot \frac{\Gamma^{t-1}(v)}{\Gamma^{t-1}(u)}, \quad \text{for all } e = (u, v) \in E(G^t). \tag{5 revisited.}$$

On the other hand, in the Hedge algorithm, the path probabilities are updated as follows: Given a learning rate $\eta$, define $P_h^1(\mathfrak{p}) = \prod_{e \in \mathfrak{p}} \phi^1(e)$, and for $t \geq 2$,

$$P_h^t(\mathfrak{p}) = \frac{P_h^{t-1}(\mathfrak{p}) \exp(\eta \sum_{e \in \mathfrak{p}} w^{t-1}(e))}{\sum_{\mathfrak{q}} P_h^{t-1}(\mathfrak{q}) \exp(\eta \sum_{e \in \mathfrak{q}} w^{t-1}(e))}. \tag{23}$$

The subscript '$h$' in $P_h^t(\mathfrak{p})$ stands for Hedge. To show that the update rule in Algorithm 2 is equivalent to the one in Hedge, we first prove the following statement:

(†) For any vertex $u$ in the graph, let $\mathcal{P}(u)$ be the set of all paths from $u$ to $z_+$. Then

$$\Gamma^{t-1}(u) = \sum_{\mathfrak{q} \in \mathcal{P}(u)} \prod_{e \in \mathfrak{q}} \left[ \phi^{t-1}(e) \exp(\eta w^{t-1}(e)) \right]. \tag{24}$$

We prove equation (24) by induction on $u$, from the bottom layer to the top layer. If $u = z_+$, by definition $\Gamma^{t-1}(z_+) = 1$, and (24) holds. Now suppose that (24) holds for all $u$ in the $(k+1)$-st layer, for some $0 \le k \le K$. Then if $u$ is in the $k$-th layer, the recursion of $\Gamma^{t-1}$ gives that

$$
\begin{aligned}
\Gamma^{t-1}(u) &= \sum_{v:(u,v)\in E} \phi^{t-1}((u,v)) \exp(\eta w^{t-1}((u,v)) \cdot \Gamma^{t-1}(v) \\
&= \sum_{v:(u,v)\in E} \phi^{t-1}((u,v)) \exp(\eta w^{t-1}((u,v))) \cdot \sum_{\mathfrak{q}\in\mathcal{P}(v)} \prod_{e\in\mathfrak{q}} \left[\phi^{t-1}(e) \exp(\eta w^{t-1}(e))\right] \\
&= \sum_{\mathfrak{q}\in\mathcal{P}(u)} \prod_{e\in\mathfrak{q}} \left[\phi^{t-1}(e) \exp(\eta w^{t-1}(e))\right],
\end{aligned}
$$

and therefore (24) holds.

Having shown equation (24), we prove by induction on $t$ our claim that Algorithm 2 is a correct implementation of Hedge, i.e. that $P^t(\mathfrak{p}) = P_h^t(\mathfrak{p})$ for all paths $\mathfrak{p}$ and rounds $t$. For $t = 1$, by definition of our initialization we have $P_h^1(\mathfrak{p}) = P^1(\mathfrak{p})$. Suppose that at time $t-1$, for every path $\mathfrak{p}$ we have $P_h^{t-1}(\mathfrak{p}) = P^{t-1}(\mathfrak{p}) = \prod_{e\in\mathfrak{p}} \phi^{t-1}(e)$. Then at time $t$, it holds that

$$
\begin{aligned}
P^t(\mathfrak{p}) = \prod_{e\in\mathfrak{p}} \phi^t(e) &\overset{(a)}{=} \prod_{e=(u,v)\in\mathfrak{p}} \left[\phi^{t-1}(e) \cdot \exp\left(\eta \cdot w^{t-1}(e)\right) \cdot \frac{\Gamma^{t-1}(v)}{\Gamma^{t-1}(u)}\right] \\
&\overset{(b)}{=} P_h^{t-1}(\mathfrak{p}) \exp\left(\eta \sum_{e\in\mathfrak{p}} w^{t-1}(e)\right) \cdot \frac{\Gamma^{t-1}(z_+)}{\Gamma^{t-1}(z_-)},
\end{aligned}
$$

where step (a) is due to equation (5), and step (b) follows using telescoping and the induction hypothesis $P_h^{t-1}(\mathfrak{p}) = \prod_{e\in\mathfrak{p}} \phi^{t-1}(e)$.

Given equation (23), by applying equation (24) to $u \in \{z_-, z_+\}$ and the induction hypothesis $P_h^{t-1}(\mathfrak{p}) = \prod_{e\in\mathfrak{p}} \phi^{t-1}(e)$, the induction is complete.

Thus Algorithm 2 is a correct implementation of Hedge.

**Step II: Regret upper bound.** Let

$$
\varepsilon = v_{i,1}\sqrt{K/T}\,. \tag{25}
$$

Applying Lemma 3 yields

$$
\mathrm{Reg}_i(\pi_i, H_{-i}^T) \le \mathrm{Reg}_i(\pi_i, H_{-i}^T, \varepsilon) + TK\varepsilon = \mathrm{Reg}_i(\pi_i, H_{-i}^T, \varepsilon) + v_{i,1}\sqrt{TK^3}, \tag{26}
$$

where $\mathrm{Reg}_i(\pi_i, H_{-i}^T, \varepsilon)$ is the counterpart of (3) where $\mathcal{S}_i$ is replaced by the set

$$
\mathcal{S}_\varepsilon = \{\varepsilon, 2\varepsilon, \cdots, \lceil v_{i,1}/\varepsilon\rceil\varepsilon\}\,.
$$

We also claim that $\sum_{e\in\mathfrak{p}} w^t(e) \le Kv_{i,1}$ for each path $\mathfrak{p}$ from source to sink in $G^t$. To see this, let $\boldsymbol{\beta}$ be the bid vector corresponding to path $\mathfrak{p}$. As shown in Lemma 1 and using the fact that edges outgoing from $z_-$ have weight zero, we have $u_i(\boldsymbol{\beta}, b_{-i}^t) = \sum_{e\in\mathfrak{p}} w^t(e)$. Since player $i$'s utility satisfies $u_i(\boldsymbol{\beta}, \mathbf{b}_{-i}^t) \le V_i(x_i(\boldsymbol{\beta}, \mathbf{b}_{-i}^t)) \le Kv_{i,1}$, we obtain $\sum_{e\in\mathfrak{p}} w^t(e) \le Kv_{i,1}$.

By Step I, Algorithm 2 is a correct implementation of Hedge. To bound the regret of the algorithm, we will invoke Corollary 1—which is a slight variant of [CBL06, Theorem 2.2]—with the following parameters:

- $N$ experts, where each expert is a path from source to sink in $G^t$;

- learning rate $\eta$, time horizon $T$, and maximum reward $L = Kv_{i,1}$;

- initial distribution $\sigma$ on the experts, where $\sigma_{\mathfrak{p}} = P^1(\mathfrak{p})$ for each expert (path from source to sink) $\mathfrak{p}$.

By Corollary 1, we obtain

$$\text{Reg}_i(\pi_i, H_{-i}^T, \varepsilon) \leq \frac{1}{\eta} \max_{\mathfrak{p} \in [N]} \log\left(\frac{1}{\sigma_{\mathfrak{p}}}\right) + \frac{TL^2\eta}{8}. \tag{27}$$

Recall Algorithm 2 initially selects a path by starting at the source $z_-$ and then performing an unbiased random walk in the layered DAG until reaching the sink $z_+$. Since the number of vertices in each layer is $\lceil v_{i,1}/\varepsilon \rceil$, the initial probability of selecting a particular expert (i.e. path from source to sink) $\mathfrak{p}$ is

$$\sigma_{\mathfrak{p}} = P^1(\mathfrak{p}) \geq \frac{1}{\lceil v_{i,1}/\varepsilon \rceil^K}, \quad \forall \mathfrak{p} \in [N]. \tag{28}$$

Substituting (28) in (27) yields

$$\text{Reg}_i(\pi_i, H_{-i}^T, \varepsilon) \leq \frac{1}{\eta} \max_{\mathfrak{p} \in [N]} \log\left(\frac{1}{\sigma_{\mathfrak{p}}}\right) + \frac{TL^2\eta}{8} \leq \frac{K \log\left(\lceil v_{i,1}/\varepsilon \rceil\right)}{\eta} + \frac{TL^2\eta}{8}$$

$$= \frac{K \log\left(\left\lceil \sqrt{\frac{T}{K}} \right\rceil\right)}{\eta} + \frac{T\left(K v_{i,1}\right)^2 \eta}{8} \quad \text{(Since } \epsilon = v_{i,1}\sqrt{\frac{K}{T}} \text{ and } L = K v_{i,1}.\text{)}$$

$$\leq \frac{K \log T}{\eta} + \frac{T\left(K v_{i,1}\right)^2 \eta}{8}. \tag{29}$$

For $\eta = \sqrt{\log T}/(v_{i,1}\sqrt{KT})$, inequality (29) gives

$$\text{Reg}_i(\pi_i, H_{-i}^T, \varepsilon) \leq \frac{K \cdot \log T}{\sqrt{\log T}/(v_{i,1}\sqrt{KT})} + \frac{T\sqrt{\log T}}{8(v_{i,1}\sqrt{KT})} \cdot \left(K v_{i,1}\right)^2$$

$$= \frac{9\, v_{i,1}}{8} \sqrt{TK^3 \log T}. \tag{30}$$

Combining inequalities (26) and (30), we get that the regret of player $i$ when running Algorithm 2 is

$$\text{Reg}_i(\pi_i, H_{-i}^T) \leq \text{Reg}_i(\pi_i, H_{-i}^T, \varepsilon) + v_{i,1}\sqrt{TK^3}$$

$$\leq \frac{9\, v_{i,1}}{8} \sqrt{TK^3 \log T} + v_{i,1}\sqrt{TK^3} \in O\left(v_{i,1}\sqrt{TK^3 \log T}\right). \tag{31}$$

This completes the proof of the regret upper bound.

**Polynomial time implementation.** Finally we analyze the running time of the above algorithm. At every step, the computation of path kernels traverses all edges (note that each edge only appears in the sum once) and could be done in $O(|E|) = O(K v_{i,1}^2/\varepsilon^2)$ time. Similarly, the update of edge probabilities $\phi^t$ for all edges also takes $O(|E|) = O(K v_{i,1}^2/\varepsilon^2)$ time.

Therefore, the overall computational complexity is $O(TK v_{i,1}^2/\varepsilon^2) = O(T^2)$, where we used the choice of $\varepsilon = v_{i,1}\sqrt{K/T}$. This completes the proof of the theorem. $\qquad \square$

## B.2 Bandit Feedback

In this section we include the main theorem and proof for the bandit setting. Recall that our algorithm for the bandit setting is the same as Algorithm 2, only with $w^t(e)$ replaced by $\widehat{w}^t(e)$:

$$\widehat{w}^t(e) = \overline{w}(e) - \frac{\overline{w}(e) - w^t(e)}{p^t(e)} \mathbb{1}_{\{e \in \mathfrak{p}^t\}}, \qquad \text{with } p^t(e) = \sum_{\mathfrak{p}: e \in \mathfrak{p}} P^t(\mathfrak{p}),$$

with $P^t$ given in (22), and

$$\overline{w}(e) = \begin{cases} v_{i,1} - r + j(r - s) & \text{if } e = (z_{r,j}, z_{s,j+1}), \\ v_{i,1} - r + Kr & \text{if } e = (z_{r,K}, z_+), \\ 0 & \text{if } e = (z_-, z_{r,1}). \end{cases}$$

The resolution parameter and learning rate are chosen to be

$$\varepsilon = v_{i,1} \min\{(K^3 \log T/T)^{1/4}, 1\}, \quad \eta = \min\left\{\varepsilon\sqrt{\log(v_{i,1}/\varepsilon)/(TK^3 v_{i,1}^4)}, 1/(Kv_{i,1})\right\}. \quad (32)$$

**Theorem 3 (restated).** *For each player $i$ and time horizon $T$, under the bandit feedback, there is an algorithm for bidding that runs in time $O(TK + K^{-5/4}T^{7/4})$ and guarantees the player's regret is at most $O(\min\{v_{i,1}(T^3 K^7 \log T)^{1/4}, v_{i,1}KT\})$.*

*Proof.* By Lemma 3 and the choice of $\varepsilon$ in (32), it suffices to show that the above algorithm $\pi_i$ runs in time $O(TKv_{i,1}^3/\varepsilon^3)$ and achieves

$$\mathrm{Reg}_i(\pi_i, H_{-i}^T, \varepsilon) = O\left(v_{i,1}^2 \sqrt{TK^5 \log(v_{i,1}/\varepsilon)}/\varepsilon + v_{i,1}K^2 \log(v_{i,1}/\varepsilon)\right).$$

The claimed result then follows from the fact that the regret is always upper bounded by $O(v_{i,1}KT)$.

Before we proceed to the proof, we first comment on the choice of estimator $\widehat{w}^t(e)$. First of all, this is an unbiased estimator of $w^t(e)$, i.e. $\mathbb{E}_{\mathfrak{p}^t \sim P^t}[\widehat{w}^t(e)] = w^t(e)$ for every edge $e$ in $G^t$. Second, instead of using the natural importance-weighted estimator $\widehat{w}^t(e) = w^t(e)\mathbb{1}_{\{e \in \mathfrak{p}^t\}}/p^t(e)$, the current form in (6) is the loss-based importance-weighted estimator used for technical reasons, similar to [LS20, Eqn. (11.6)]. Third, by exploiting our DAG structure and the definition of $w^t(e)$ in (4), we construct an edge-specific quantity $\overline{w}(e)$ which always upper bounds $w^t(e)$.

We now analyze the regret of the algorithm with $\widehat{w}^t(e)$ given by (6). The standard EXP3 analysis (see, e.g. [LS20, Chapter 11]) gives that

$$\mathrm{Reg}_i(\pi_i, H_{-i}^T, \varepsilon) \leq \frac{1}{\eta} \max_{\mathfrak{p}} \log \frac{1}{P^1(\mathfrak{p})} + \sum_{t=1}^{T} \mathbb{E}\left[\frac{1}{\eta} \log\left(\sum_{\mathfrak{p}} P^t(\mathfrak{p})e^{\eta\widehat{w}^t(\mathfrak{p})}\right) - \sum_{\mathfrak{p}} P^t(\mathfrak{p})\widehat{w}^t(\mathfrak{p})\right], \quad (33)$$

where $\widehat{w}^t(\mathfrak{p}) = \sum_{e \in \mathfrak{p}} \widehat{w}^t(e)$ is the estimated total weight of path $\mathfrak{p}$, and the expectation is with respect to the randomness in the estimator $\widehat{w}^t(e)$. For every path $\mathfrak{p} = (z_-, z_{r_1,1}, \cdots, z_{r_K,K}, z_+)$ from the source to the sink, we have (by convention $r_{K+1} = 0$):

$$\widehat{w}^t(\mathfrak{p}) = \sum_{e \in \mathfrak{p}} w^t(e) \leq \sum_{e \in \mathfrak{p}} \overline{w}(e) = \sum_{j=1}^{K}(v_{i,1} - (j-1)r_j + jr_{j+1}) = Kv_{i,1}.$$

Since $e^x \leq 1 + x + x^2$ whenever $x \leq 1$ and $\log(1+y) \leq y$ whenever $y > -1$, if $\eta \leq 1/(Kv_{i,1})$, inequality (33) gives

$$\mathrm{Reg}_i(\pi_i, H_{-i}^T, \varepsilon) \leq \frac{1}{\eta} \max_{\mathfrak{p}} \log \frac{1}{P^1(\mathfrak{p})} + \eta \sum_{t=1}^{T} \sum_{\mathfrak{p}} P^t(\mathfrak{p}) \, \mathbb{E}[\widehat{w}^t(\mathfrak{p})^2]$$

$$\leq \frac{K}{\eta} \log\left\lceil \frac{v_{i,1}}{\varepsilon} \right\rceil + \eta \sum_{t=1}^{T} \sum_{\mathfrak{p}} P^t(\mathfrak{p}) \cdot (K+1) \sum_{e \in \mathfrak{p}} \mathbb{E}[\widehat{w}^t(e)^2],$$

where the last equality uses $(\sum_{i=1}^{n} x_i)^2 \leq n \sum_{i=1}^{n} x_i^2$, and that each path $\mathfrak{p}$ from the source to the sink has length $K+1$. To proceed, note that

$$\mathbb{E}[\widehat{w}^t(e)^2] = \overline{w}(e)^2 \cdot (1 - p^t(e)) + \left(\overline{w}(e) - \frac{\overline{w}(e) - w^t(e)}{p^t(e)}\right)^2 \cdot p^t(e)$$

$$= w^t(e)^2 + (\overline{w}(e) - w^t(e))^2 \cdot \left(\frac{1}{p^t(e)} - 1\right),$$

and therefore

$$\sum_{\mathfrak{p}} P^t(\mathfrak{p}) \sum_{e \in \mathfrak{p}} \mathbb{E}[\widehat{w}^t(e)^2] = \sum_{\mathfrak{p}} P^t(\mathfrak{p}) \sum_{e \in \mathfrak{p}} \left[w^t(e)^2 + (\overline{w}(e) - w^t(e))^2 \cdot \left(\frac{1}{p^t(e)} - 1\right)\right]$$

$$= \sum_{e} \left[w^t(e)^2 + (\overline{w}(e) - w^t(e))^2 \cdot \left(\frac{1}{p^t(e)} - 1\right)\right] \sum_{\mathfrak{p}:e \in \mathfrak{p}} P^t(\mathfrak{p})$$

$$= \sum_{e} \left[w^t(e)^2 p^t(e) + (\overline{w}(e) - w^t(e))^2(1 - p^t(e))\right] \leq \sum_{e} \overline{w}(e)^2,$$

where in the middle we have used the definition $\sum_{\mathfrak{p}:e\in\mathfrak{p}} P^t(\mathfrak{p}) = p^t(e)$ in (6). To further upper bound the above quantity, note that

$$
\begin{aligned}
\sum_e \overline{w}(e)^2 &\le \sum_{j=1}^K \sum_{1\le s\le r\le \lceil v_{i,1}/\varepsilon\rceil} (v_{i,1} + (j-1)r\varepsilon)^2 \\
&\le \left\lceil \frac{v_{i,1}}{\varepsilon}\right\rceil \sum_{j=1}^K \sum_{r=1}^{\lceil v_{i,1}/\varepsilon\rceil} (V + (j-1)r\varepsilon)^2 \\
&\le 2\left\lceil \frac{v_{i,1}}{\varepsilon}\right\rceil \sum_{j=1}^K \sum_{r=1}^{\lceil v_{i,1}/\varepsilon\rceil} (v_{i,1}^2 + (j-1)^2 r^2\varepsilon^2) = O\left(\frac{K^3 v_{i,1}^4}{\varepsilon^2}\right).
\end{aligned}
$$

A combination of the above inequalities shows that as long as $\eta \le 1/(Kv_{i,1})$,

$$
\text{Reg}_i(\pi_i, H_{-i}^T, \varepsilon) = O\left(\frac{K}{\eta}\log\frac{v_{i,1}}{\varepsilon} + \frac{\eta T K^4 v_{i,1}^4}{\varepsilon^2}\right).
$$

Next, by the choice of $\eta$ in (32), we have

$$
\text{Reg}_i(\pi_i, H_{-i}^T, \varepsilon) = O\left(\frac{v_{i,1}^2\sqrt{TK^5\log(v_{i,1}/\varepsilon)}}{\varepsilon} + K^2 v_{i,1}\log\frac{v_{i,1}}{\varepsilon}\right).
$$

As for the computational complexity, the only difference from the Algorithm 2 is the additional computation of the estimated weights $\widehat{w}^t(e)$ in (6), or equivalently, the marginal probability $p^t(e)$. As $P^t$ has a product structure in (22), the celebrated message passing algorithm in graphical models [WJ08, Section 2.5.1] takes $O(|E|v_{i,1}/\varepsilon) = O(Kv_{i,1}^3/\varepsilon^3)$ time to compute all edge marginals $\{p^t(e)\}_{e\in E}$. Therefore the overall computational complexity is $O(TKv_{i,1}^3/\varepsilon^3)$. $\qquad\square$

## B.3 Regret Lower Bound

In this section we include the theorem and proof for the regret lower bound. The construction uses the $(K+1)$-st highest price, but is similar for the $K$-th highest price.

**Theorem 4 (restated, formal).** *Let $K \ge 2$. For any policy $\pi_i$ used by player $i$, there exists a bid sequence $\{\mathbf{b}_{-i}^t\}_{t=1}^T$ for the other players such that the expected regret in (3) satisfies $\mathbb{E}[\text{Reg}_i(\pi_i, \{\mathbf{b}_{-i}^t\}_{t=1}^T)] \ge cv_{i,1}K\sqrt{T}$, where $c > 0$ is an absolute constant.*

*Proof.* Without loss of generality assume that $K = 2k$ is an even integer, and by scaling we may assume that $v_{i,1} = 1$. Consider the following two scenarios:

- the utility of the bidder $i$ is $v_{i,j} \equiv 1$, for all $j \in [K]$;

- at scenario 1, for every $t \in [T]$, the other bidders' bids are

$$
\mathbf{b}_{-i}^t = \begin{cases} \left(\frac{2}{3}, \frac{2}{3}, \cdots, \frac{2}{3}, 0, \cdots, 0\right) & \text{with probability } 0.5 + \delta, \\ \left(\frac{2}{3}, \frac{2}{3}, \cdots, \frac{2}{3}, \frac{2}{3}, \cdots, \frac{2}{3}\right) & \text{with probability } 0.5 - \delta, \end{cases}
$$

where the number of non-zero entries is $k$ in the first line and $2k$ in the second line, and $\delta \in (0, 1/4)$ is a parameter to be determined later. The randomness used at different times is independent.

- at scenario 2, for every $t \in [T]$, the other bidders' bids are

$$
\mathbf{b}_{-i}^t = \begin{cases} \left(\frac{2}{3}, \frac{2}{3}, \cdots, \frac{2}{3}, 0, \cdots, 0\right) & \text{with probability } 0.5 - \delta, \\ \left(\frac{2}{3}, \frac{2}{3}, \cdots, \frac{2}{3}, \frac{2}{3}, \cdots, \frac{2}{3}\right) & \text{with probability } 0.5 + \delta, \end{cases}
$$

where the number of non-zero entries is $k$ in the first line and $2k$ in the second line, and $\delta \in (0, 1/4)$ is a parameter to be determined later. The randomness used at different times is independent.

We denote by $P$ and $Q$ the distributions of $\{\mathbf{b}_{-i}^t\}_{t=1}^T$ under scenarios 1 and 2, respectively, then

$$
\begin{aligned}
D_{\mathrm{KL}}(P\|Q) &= T \cdot D_{\mathrm{KL}}(\mathrm{Bern}(0.5+\delta)\|\mathrm{Bern}(0.5-\delta)) \\
&\leq T \cdot \chi^2(\mathrm{Bern}(0.5+\delta)\|\mathrm{Bern}(0.5-\delta)) \\
&= T \cdot \frac{(2\delta)^2}{(0.5+\delta)(0.5-\delta)} \leq \frac{64}{3}T\delta^2.
\end{aligned}
$$

Consequently, by [Tsy09, Lemma 2.6],

$$
1 - \mathrm{TV}(P,Q) \geq \frac{1}{2}\exp\left(-D_{\mathrm{KL}}(P\|Q)\right) \geq \frac{1}{2}\exp\left(-\frac{64T\delta^2}{3}\right).
$$

Next we investigate the separation between these two scenarios. It is clear that

$$
\begin{aligned}
\max_{\mathbf{b}_i} \mathbb{E}_P\left[u_i(\mathbf{b}_i;\mathbf{b}_{-i}^t)\right] &\geq \mathbb{E}_P\left[u_i((1,1,\cdots,1,0,\cdots,0);\mathbf{b}_{-i}^t)\right] \\
&= \left(\frac{1}{2}+\delta\right)\cdot k + \left(\frac{1}{2}-\delta\right)\cdot\frac{k}{3} = \frac{2+2\delta}{3}\cdot k; \\
\max_{\mathbf{b}_i} \mathbb{E}_Q\left[u_i(\mathbf{b}_i;\mathbf{b}_{-i}^t)\right] &\geq \mathbb{E}_Q\left[u_i((1,1,\cdots,1,1,\cdots,1);\mathbf{b}_{-i}^t)\right] \\
&= \left(\frac{1}{2}-\delta\right)\cdot\frac{2k}{3} + \left(\frac{1}{2}+\delta\right)\cdot\frac{2k}{3} = \frac{2k}{3}.
\end{aligned}
$$

Moreover, under $(P+Q)/2$ (i.e. $\mathbf{b}_{-i}^t$ follows a $\mathrm{Bern}(1/2)$ distribution), suppose that the vector $\mathbf{b}_i$ has $k'$ components smaller than $2/3$. Distinguish into two scenarios:

- if $k' < k$, then

$$
\mathbb{E}_{(P+Q)/2}\left[u_i(\mathbf{b}_i;\mathbf{b}_{-i}^t)\right] \leq \frac{1}{2}\cdot\frac{2k-k'}{3} + \frac{1}{2}\cdot\frac{2k-k'}{3} \leq \frac{2k}{3};
$$

- if $k' \geq k$, then

$$
\mathbb{E}_{(P+Q)/2}\left[u_i(\mathbf{b}_i;\mathbf{b}_{-i}^t)\right] \leq \frac{1}{2}\cdot(2k-k') + \frac{1}{2}\cdot\frac{2k-k'}{3} \leq \frac{2k}{3}.
$$

Therefore, it always holds that

$$
\max_{\mathbf{b}_i}\mathbb{E}_{(P+Q)/2}\left[u_i(\mathbf{b}_i;\mathbf{b}_{-i}^t)\right] \leq \frac{2k}{3}.
$$

Consequently, for each $\mathbf{b}_i$,

$$
\begin{aligned}
&\max_{\mathbf{b}_i^\star}\mathbb{E}_P\left[u_i(\mathbf{b}_i^\star;\mathbf{b}_{-i}^t) - u_i(\mathbf{b}_i;\mathbf{b}_{-i}^t)\right] + \max_{\mathbf{b}_i^\star}\mathbb{E}_Q\left[u_i(\mathbf{b}_i^\star;\mathbf{b}_{-i}^t) - u_i(\mathbf{b}_i;\mathbf{b}_{-i}^t)\right] \\
&\geq \max_{\mathbf{b}_i^\star}\mathbb{E}_P\left[u_i(\mathbf{b}_i^\star;\mathbf{b}_{-i}^t)\right] + \max_{\mathbf{b}_i^\star}\mathbb{E}_Q\left[u_i(\mathbf{b}_i^\star;\mathbf{b}_{-i}^t)\right] - 2\max_{\mathbf{b}_i}\mathbb{E}_{(P+Q)/2}\left[u_i(\mathbf{b}_i;\mathbf{b}_{-i}^t)\right] \\
&\geq \frac{2+2\delta}{3}k + \frac{2k}{3} - 2\cdot\frac{2k}{3} = \frac{2\delta k}{3}.
\end{aligned}
$$

In other words, any bid vector $\mathbf{b}_i$ either incurs a total regret $(\delta kT)/3$ under $P$, or incurs a total regret $(\delta kT)/3$ under $Q$.

Now the classical two-point method (see, e.g. [Tsy09, Theorem 2.2]) gives

$$
\mathbb{E}_{(P+Q)/2}[\mathrm{Reg}_i(\pi_i,\{\mathbf{b}_{-i}^t\}_{t=1}^T)] \geq \frac{\delta kT}{3}\cdot(1-\mathrm{TV}(P,Q)) \geq \frac{\delta kT}{6}\exp\left(-\frac{64T\delta^2}{3}\right)
$$

for every $\delta \in (0,1/4)$. Choosing $\delta = 1/(8\sqrt{T})$ gives that

$$
\mathbb{E}_{(P+Q)/2}[\mathrm{Reg}_i(\pi_i,\{\mathbf{b}_{-i}^t\}_{t=1}^T)] \geq \frac{K\sqrt{T}}{96e^{1/3}},
$$

i.e. the claimed lower bound holds with $c = 1/(96e^{1/3})$. $\qquad\square$

# C Appendix: Equilibrium Analysis

In this section we show that the pure Nash equilibria with price zero of the $(K+1)$-st auction are the only ones that are robust to deviations by groups of players, captured through the notion of the core in the game among the bidders. The core of a game was formulated by Edgeworth [Edg81] and brought into game theory by Gillies [Gil59]. There is an extensive body of literature on the core of various games, including for auctions; see, e.g., analysis of collusion (cartel behavior) in first price auctions in [Pes00].

Our focus here is the core of the game among the bidders, where the auctioneer first sets the auction format and then the bidders can strategize and collude among themselves, without the auctioneer. Roughly speaking, a strategy profile is core-stable if no group $C$ of players can deviate simultaneously (i.e. each player $i \in C$ deviates to some alternative strategy profile), such that each player in $C$ weakly improves and the improvement is strict for at least one player. For a deviation to take place, the players in $C$ coordinate and switch their strategies simultaneously, while the players outside $C$ keep their previous strategies. Every core-stable strategy profile is also a Nash equilibrium, since a strategy profile that is stable against deviations by groups of players is also stable against deviations by individuals.

We consider two variants of the core, with and without monetary transfers [SS69, Bon63, Sha67]. In the case with transfers, the players can make monetary payments to each other, and so a strategy profile consists of a tuple of bids and transfers. In the case without transfers, the players cannot make such transfers and their strategy is the bid vector; thus the only agreement they can make in this case is to coordinate their bids.

## C.1 Core with transfers

A strategy profile in this setting is described by a tuple $(\mathbf{b}, \mathbf{t})$, where $\mathbf{b}$ is a bid profile and $\mathbf{t}$ is a profile of payments (aka monetary transfers), such that $t_{i,j} \geq 0$ is the monetary payment of player $i$ to player $j$. At this strategy profile, the auctioneer runs the auction with bids $\mathbf{b}$ and returns the outcome (price and allocation), while the players make the monetary transfers $\mathbf{t}$ to each other.

For each profile of monetary transfers $\mathbf{t}$, let $m_i(\mathbf{t})$ be the net amount of money that player $i$ gets after all the transfers are made:

$$m_i(\mathbf{t}) = \sum_{j=1}^{n} t_{j,i} - \sum_{j=1}^{n} t_{i,j} \,.$$

The utility of player $i$ at profile $(\mathbf{b}, \mathbf{t})$ is

$$u_i(\mathbf{b}, \mathbf{t}) = m_i(\mathbf{t}) + \left( \sum_{j=1}^{x_i(\mathbf{b})} v_{i,j} \right) - p \cdot x_i(\mathbf{b}) \,.$$

**Deviations.** Since in the case of auctions the actions (e.g. bids) of a group of players can affect the utility of the players outside of the group, it is necessary to model how the players outside $S$ react to the deviation. Such reactions have been studied in the literature on the core with externalities (see, e.g., [Koc07, Koc09]).

We consider neutral reactions, where non-deviators (i.e. players outside $S$) have a mild reaction to the deviation: they maintain the same bids as before the deviation and the monetary transfer of each player $i \in [n] \setminus S$ to each player $j \in S$ is non-negative. The core where the deviators assume the non-deviators will have neutral reactions to the deviation is known as the neutral core [SMR$^+$13]. Several other variants of the core exist, such as pessimistic core, where each deviator assumes that they will be punished in the worst possible way by the non-deviators. Such variants are also interesting to study, but next we focus on the basic case of neutral reactions.

A group of players will alternatively be called a *coalition*. The set of all players, $[n]$, will also be called sometimes the grand coalition.

A group of players that agree on a deviation are known as a blocking coalition, formally defined next.

**Definition 3** (Blocking coalition, with transfers). *Let* $(\mathbf{b}, \mathbf{t})$ *be a tuple of bids and monetary transfers. A group $S \subseteq [n]$ of players is a* blocking coalition *if there exists a profile* $(\widetilde{\mathbf{b}}, \widetilde{\mathbf{t}})$, *at which each player $i \in S$ weakly improves their utility, the improvement is strict for at least one player in $S$, and*

- $\widetilde{b}_{i,j} = b_{i,j}$ *if* $i \in [n] \setminus S, j \in [K]$.

- $\widetilde{t}_{i,j} = 0$ *if* $i \in [n] \setminus S$ *and* $j \in S$.

In other words, the blocking coalition $S$ needs to agree on their bids and transfers to each other, such that they improve their utility when the players outside $S$ maintain their existing bids but stop payments to players in $S$. In fact our characterization holds even if the players outside $S$ make any non-negative transfers to the players in $S$; the case where the transfers are zero is the extreme case. If a coalition $S$ deviates with zero transfers from players outside $S$, it also deviates for any transfers that are non-negative.

**Definition 4** (The core with transfers). *The core with transfers* *consists of profiles* $(\mathbf{b}, \mathbf{t})$ *at which there are no blocking coalitions. Such profiles are* core-stable.

**Theorem 6 (Core with transfers; restated).** *Consider $K$ units and $n > K$ hungry players. The core with transfers of the $(K + 1)$-st auction can be characterized as follows:*

- *Let $(\mathbf{b}, \mathbf{t})$ be an arbitrary tuple of bids and transfers that is core stable. Then the allocation $\mathbf{x}(\mathbf{b})$ maximizes social welfare, the price is zero (i.e. $p(\mathbf{b}) = 0$), and there are no transfers between the players (i.e. $\mathbf{t} = 0$).*

*Proof.* The proof has three steps as follows.

**Step I: core transfers are zero.** Assume towards a contradiction that $(\mathbf{b}, \mathbf{t})$ is core stable and $\mathbf{t} \neq 0$. Consider the directed weighted graph $G = ([n], E, \mathbf{t})$, where $E$ consists of all the directed edges $(i, j)$ and the weight of each edge is $t_{i,j}$. The net amount of money that each player $i$ gets from transfers is $m_i(\mathbf{t}) = \sum_{j=1}^{n} t_{j,i} - \sum_{j=1}^{n} t_{i,j}$.

If there is a cycle $C = (i_1, \ldots, i_k)$ such that the payments along the cycle are strictly positive: $t_{i_1,i_2} > 0, \ldots, t_{i_{k-1},i_k} > 0$, and $t_{i_k,i_1} > 0$, then some cancellations take place. That is, by subtracting $\min\{t_{i_1,i_2}, \ldots, t_{i_{k-1},i_k}, t_{i_k,i_1}\}$ from the weight of each edge $(i, j) \in C$, we obtain a weighted directed graph without cycle $C$ and where each player has the same net amount of money as in the original graph. Iterating the operation of removing cycles, we obtain a directed acyclic graph where the players have the same net amount of money as in the original graph.

Thus we can in fact assume the transfers $\mathbf{t}$ are such that $G$ is acyclic.

Consider a topological ordering $(i_1, \ldots, i_n)$ of the vertices of $G$. Let $j$ be the minimum index for which vertex $i_j$ has no incoming edges and at least one outgoing edge. Then $m_{i_j}(\mathbf{t}) < 0$. The utility of player $i_j$ can be upper bounded as follows:

$$u_{i_j}(\mathbf{b}, \mathbf{t}) = m_{i_j}(\mathbf{t}) + \left( \sum_{k=1}^{x_{i_j}(\mathbf{b})} v_{i_j,k} \right) - p(\mathbf{b}) \cdot x_{i_j}(\mathbf{b}) < \left( \sum_{k=1}^{x_{i_j}(\mathbf{b})} v_{i_j,k} \right) - p(\mathbf{b}) \cdot x_{i_j}(\mathbf{b}).$$

We claim that player $i_j$ has an improving deviation by keeping its bid vector $\mathbf{b}_{i_j}$ and stopping all payments to other players. Since the other players have neutral reactions to the deviations, we obtain an outcome $(\widetilde{\mathbf{b}}, \widetilde{\mathbf{t}})$ such that $\widetilde{\mathbf{b}} = \mathbf{b}$, $\widetilde{t}_{i_j,k} = 0$ for all $k \in [n]$, and $\widetilde{t}_{k,i_j} \leq t_{k,i_j}$ for all $k \neq i_j$. The gain of player $i_j$ from the deviation can be bounded by:

$$u_{i_j}(\widetilde{\mathbf{b}}, \widetilde{\mathbf{t}}) - u_{i_j}(\mathbf{b}, \mathbf{t}) = u_{i_j}(\mathbf{b}, \widetilde{\mathbf{t}}) - u_{i_j}(\mathbf{b}, \mathbf{t}) \qquad \text{(Since } \widetilde{\mathbf{b}} = \mathbf{b}.)$$

$$= \left[ m_{i_j}(\widetilde{\mathbf{t}}) + \left( \sum_{k=1}^{x_{i_j}(\mathbf{b})} v_{i_j,k} \right) - p(\mathbf{b}) \cdot x_{i_j}(\mathbf{b}) \right] - \left[ m_{i_j}(\mathbf{t}) + \left( \sum_{k=1}^{x_{i_j}(\mathbf{b})} v_{i_j,k} \right) - p(\mathbf{b}) \cdot x_{i_j}(\mathbf{b}) \right]$$

$$= m_{i_j}(\widetilde{\mathbf{t}}) - m_{i_j}(\mathbf{t})$$

$$= -m_{i_j}(\mathbf{t}) \qquad \text{(Since } m_{i_j}(\widetilde{\mathbf{t}}) = 0.)$$

$$> 0. \qquad \text{(Since } m_{i_j}(\mathbf{t}) < 0 \text{ by choice of } i_j.)$$

Thus player $i_j$ has a strictly improving deviation, which contradicts the choice of $(\mathbf{b}, \mathbf{t})$ as core-stable with $\mathbf{t} \neq \mathbf{0}$. Thus the assumption must have been false. It follows that the only core stable profiles (if any) have $\mathbf{t} = \mathbf{0}$.

**Step II: the social welfare is maximized.** Next we show that if $(\mathbf{b}, \mathbf{t})$ is a core-stable outcome, then the allocation induced by $\mathbf{b}$ is welfare maximizing. Assume towards a contradiction this is not the case. By Step I, we have $\mathbf{t} = \mathbf{0}$.

Let $w_1 \geq \ldots \geq w_{n \cdot K}$ be the bids sorted in decreasing order (breaking ties lexicographically) at the truth-telling bid profile $\mathbf{v}$. For each $j \in [n \cdot K]$, let $\pi_j$ be the player that submitted bid $w_j$ in this ordering.

Let $\widetilde{w}_1 \geq \ldots \geq \widetilde{w}_{n \cdot K}$ be the bids sorted in decreasing order (breaking ties lexicographically) at the bid profile $\mathbf{b}$. Let $\widetilde{\pi}_j$ be the player that submitted bid $\widetilde{w}_j$ in this ordering.

Consider an undirected bipartite graph $G = (L, R, E)$, where $L$ is the left part, $R$ the right part, and $E$ the set of edges. Define

- $L = \{(i, j) \mid i \in [n], j \in [K], \text{ and } x_i(\mathbf{v}) \geq j\}$. For example, if $x_i(\mathbf{v}) = 2$, then $L$ has nodes $(i, 1)$ and $(i, 2)$. If on the other hand $x_i(\mathbf{v}) = 0$, then $L$ has no nodes $(i, j)$, for any $j$.

- $R = \{(i, j) \mid i \in [n], j \in [K], \text{ and } x_i(\mathbf{b}) \geq j\}$.

- $E = (s_1, s_2)$, for all $s_1 \in L$ and $s_2 \in R$.

Since both allocations $\mathbf{x}(\mathbf{v})$ and $\mathbf{x}(\mathbf{b})$ allocate exactly $K$ units, we have $|L| = |R| = K$. Consider now a graph $G_1 = (L_1, R_1, E_1)$ obtained from $G$ as follows. Set $G = G_1$. Then for each node $(i, j) \in L$: if the node also appears in $R$, then delete both copies of the node, together with any edges containing them.

Thus in $G_1$, the left side $L_1$ consists of nodes $(i, j)$ with $x_i(\mathbf{v}) \geq j$ but $x_i(\mathbf{b}) < j$. For each such node $(i, j)$, let $k$ be the rank of the valuation $v_{i,j}$ in the ordering $\pi$. By definition of $G_1$, we have that $R_1$ does not have any node of the form $(i, s)$ for any $s$. To see this, observe that

- nodes $(i, s)$ with $s < j$ were deleted when constructing $G_1$ from $G$, and

- nodes $(i, s)$ with $s \geq j$ do not exist even in $G$ (if they did, then $(i, j)$ would exist in both $L$ and $R$ and so would have been deleted when constructing $G_1$).

Let $d(i, j)$ be the player that displaces player $i$'s bid for the $j$-th unit at the bid profile $\mathbf{b}$. Formally, $d(i, j)$ is the owner of bid $\widetilde{w}_k$ when considering the bids $\mathbf{b}$ in descending order. Let $\omega(i, j) \in \mathbb{N}$ be such that player $d(i, j)$ obtains an $\omega(i, j)$-th unit in their bundle at $\mathbf{b}$.

Since $x_i(\mathbf{b}) < j$, we have $i \neq d(i, j)$. Since at the truth-telling profile player $i$ gets a $j$-th unit but player $d(i, j)$ does not get an $\omega(i, j)$-th unit, we have $v_{i,j} \geq v_{d(i,j),\omega(i,j)}$. Moreover, since the allocation $\mathbf{x}(\mathbf{v})$ maximizes welfare but $\mathbf{x}(\mathbf{b})$ does not, the inequality is strict for some $(i, j) \in L_1$.

We claim that $[n]$ is a blocking coalition. To show this, we will argue there is a bid profile $\mathbf{b}^*$ and vector of transfers $\mathbf{t}^*$ such that at $(\mathbf{b}^*, \mathbf{t}^*)$ the utility of each player $i \in [n]$ is weakly improved and the improvement is strict for at least one player. For each $i \in [n], k \in [K]$, let

$$b_{i,j}^* = \begin{cases} v_{i,j} & \text{if } x_i(\mathbf{v}) \geq j \\ 0 & \text{otherwise.} \end{cases}$$

Then $\mathbf{x}(\mathbf{b}^*) = \mathbf{x}(\mathbf{v})$ and $p(\mathbf{b}^*) = 0$. Also define monetary transfers $\mathbf{t}^*$ as follows:

- Initialize $\mathbf{t}^* = 0$. Let $\varepsilon = \min_{(i,j) \in L_1} \left( v_{i,j} - v_{d(i,j),\omega(i,j)} \right) / 2$. Then $v_{i,j} \geq v_{d(i,j),\omega(i,j)} + \varepsilon$ for each $(i, j) \in L_1$.

- For each $(i, j) \in L_1$, let $t_{i,d(i,j)}^* := t_{i,d(i,j)}^* + v_{d(i,j),\omega(i,j)} + \varepsilon$.

Thus each player $i$ that got a $j$-th unit in their bundle at the truth-telling profile did so because their bid for the $j$-th unit had rank $k \leq K$. Since $i$ does not get the $j$-th unit at bid profile $\mathbf{b}$, there is a player $d(i,j)$ whose bid for the $j$-th unit had rank $k$ and who received this way a $\omega(i,j)$-th unit in their bundle.

We argue that all the players weakly improve their utility at $(\mathbf{b}^*, \mathbf{t}^*)$, and the improvement is strict for at least one of them.

- For each pair $(i,j) \in L_1$, under the bid profile $\mathbf{b}^*$ player $i$ receives the $j$-th unit at a cost of zero and transfers an amount of $v_{d(i,j),\omega(i,j)} + \varepsilon$ to player $d(i,j)$.

  The component of the utility that player $i$ gets from unit $j$, counting the value, price, and transfer related to unit $j$, is $v_{i,j} - 0 - (v_{d(i,j),\omega(i,j)} + \varepsilon) \geq 0$, where the inequality holds by choice of $\varepsilon$. This is a weak improvement compared to the utility that player $i$ gets from unit $j$ at profile $(\mathbf{b}, \mathbf{0})$, which is zero. Moreover, the improvement is strict for at least one pair $(i,j) \in L_1$, since the bid profile $\mathbf{b}$ does not induce a welfare maximizing allocation.

- For each pair $(i,j) \in L \setminus L_1$, at the profile $(\mathbf{b}, \mathbf{0})$ player $i$ gets utility $v_{i,j} - p(\mathbf{b})$ from unit $j$, since it makes no transfers. At profile $\mathbf{b}^*$, player $i$ gets unit $j$ at a price of zero and makes no transfers (towards other players) related to unit $j$. Thus the component of the utility related to unit $j$ is $v_{i,j} - 0 - 0 = v_{i,j}$. Since $p(\mathbf{b}) \geq 0$, we have $v_{i,j} - p(\mathbf{b}) \leq v_{i,j}$, a weak improvement for player $i$ with respect to unit $j$.

- For each pair $(i,j) \notin L$: if $(i,j) \notin R$, then player $i$ does not get a $j$-th unit under either $\mathbf{b}$ or $\mathbf{b}^*$, so its utility from unit $j$ is zero at both profiles. If on the other hand $(i,j) \in R$, since $(i,j) \notin L$, it must be that $(i,j) \in R_1$. At profile $(\mathbf{b}, \mathbf{t})$ the utility of player $i$ from unit $j$ is $v_{i,j} - p(\mathbf{b})$ since it gets the unit and receives no transfers. At profile $(\mathbf{b}^*, \mathbf{t}^*)$ player $i$ does not receive the unit but receives a transfer of $v_{i,j} + \varepsilon$ from the player that gets the unit instead. This is again a weak improvement.

Thus all players weakly improve their utility at $(\mathbf{b}^*, \mathbf{t}^*)$ and the improvement is strict for at least one player, so the profile $(\mathbf{b}, \mathbf{0})$ is not stable. This is a contradiction, thus the assumption that $\mathbf{x}(\mathbf{b})$ is not welfare maximizing must have been false.

**Step III: the price is zero.** Next we show that if profile $(\mathbf{b}, \mathbf{0})$ is core-stable, then $p(\mathbf{b}) = 0$. By Step II, the bid profile $\mathbf{b}$ induces a welfare maximizing allocation.

Suppose towards a contradiction that $p(\mathbf{b}) > 0$. Then we show there exists a blocking coalition. Let $w_1 \geq \ldots \geq w_{n \cdot K}$ be the bids sorted in decreasing order (breaking ties lexicographically) at bid profile $\mathbf{b}$. For each $i \in [n \cdot K]$, let $\pi_i$ be the owner of bid $w_i$.

We match the players as follows. Create a bipartite graph with left part $L = (\pi_1, \ldots, \pi_K)$ (allowing repetitions) and right part $R = (\pi_{K+1}, \ldots, \pi_{2K})$ (allowing repetitions). For each $i \in [K]$, create edge $(\pi_i, \pi_{K+i})$. Consider the profile $(\mathbf{b}^*, \mathbf{t}^*)$, where

$$b_{i,j}^* = \begin{cases} v_{i,j} & \text{if } x_i \geq j \\ 0 & \text{otherwise.} \end{cases}$$

Define the transfers as follows:

- Initialize $\mathbf{t}^* = \mathbf{0}$. Set $\varepsilon = p/(2K)$. For each $i \in [K]$, let player $\pi_i$ pay an additional amount of $\varepsilon$ to player $\pi_{K+i}$: $t_{\pi_i, \pi_{K+i}}^* = t_{\pi_i, \pi_{K+i}}^* + \varepsilon$.

Then each player $i$ with $x_i(\mathbf{b}) > 0$ gets the same allocation at $\mathbf{b}^*$ as at $\mathbf{b}$, but pays a price of zero for the units and makes a transfer of at most $p/2$ to other players, resulting in improved utility compared to the utility at profile $(\mathbf{b}, \mathbf{0})$.

On the other hand, each player $i$ with $x_i(\mathbf{b}) = 0$ on the other hand gets the same allocation at $\mathbf{b}^*$ as at $\mathbf{b}$ (i.e. no units), but receives a non-negative amount of money from other players, and the net amount of money received is strictly positive for all the players $\pi_{K+1}, \ldots, \pi_{2K}$.

Thus there is an improving deviation, which contradicts the choice of $(\mathbf{b}, \mathbf{t})$ as core-stable. Thus the assumption must have been false, and $p(\mathbf{b}) = 0$. $\qquad \square$

## C.2 Core without transfers

A strategy profile in this setting is described by a bid profile $\mathbf{b}$, where $\mathbf{b}_i = (b_{i,1}, \ldots, b_{i,K})$ is the bid vector of player $i$. In the setting without transfers, given a profile $\mathbf{b}$ of bids, a blocking coalition $S$ needs to agree on simultaneously changing their bids, such that when the players outside $S$ still bid according to $\mathbf{b}$, the players in $S$ weakly improve their utility and the improvement is strict for at least one player in $S$.

The core stable profiles are those that have no such blocking coalitions. Formally, we have:

**Definition 5** (Blocking coalition, without transfers). *Let $\mathbf{b}$ be a bid profile. A group $S \subseteq [n]$ of players is a* blocking coalition *if there exists a bid profile $\widetilde{\mathbf{b}}$, at which each player $i \in S$ weakly improves their utility, the improvement is strict for at least one player in S, and $\widetilde{b}_{i,j} = b_{i,j}$ for all $i \in [n] \setminus S, j \in [K]$.*

**Definition 6** (The core with transfers). *The* core with transfers *consists of bid profiles $\mathbf{b}$ at which there are no blocking coalitions. Such profiles are* core-stable*.*

The non-transferable utility core can be characterized as follows.

**Theorem 5 (Core without transfers; restated).** *Consider $K$ units and $n > K$ hungry players. The core without transfers of the $(K+1)$-st auction can be characterized as follows:*

- *every bid profile $\mathbf{b}$ that is core stable has price zero (i.e. $p(\mathbf{b}) = 0$);*

- *each allocation $\mathbf{z}$ where all the units are allocated can be supported in a core stable bid profile $\mathbf{b}$ with price zero (i.e. $\mathbf{x}(\mathbf{b}) = \mathbf{z}$ and $p(\mathbf{b}) = 0$).*

*Proof.* The proof is in two parts.

**Part I: the price in the core is zero.** Consider a bid profile $\mathbf{b}$ that is core-stable. Suppose towards a contradiction that $p(\mathbf{b}) > 0$.

Let $M = \sum_{\ell_1=1}^{n} \sum_{\ell_2=1}^{K} v_{\ell_1,\ell_2}$. Define a bid profile $\widetilde{\mathbf{b}}$ such that for all $i \in [n], j \in [K]$:

$$\widetilde{b}_{i,j} = \begin{cases} M & \text{if } j \leq x_i(\mathbf{b}) \\ \varepsilon & \text{otherwise.} \end{cases} \tag{34}$$

At $\widetilde{\mathbf{b}}$, the players only submit bids equal to $M > 0$ for the units they are supposed to get at allocation $\mathbf{x}(\mathbf{b})$, and moreover, there are exactly $K$ strictly positive bids. Thus $\mathbf{x}(\widetilde{\mathbf{b}}) = \mathbf{x}(\mathbf{b})$. Moreover, $p(\widetilde{\mathbf{b}}) = 0$ since the $(K+1)$-st highest bid is 0.

Then the grand coalition $C = [n]$ is blocking with the profile $\widetilde{\mathbf{b}}$, in contradiction with $\mathbf{b}$ being core stable. Thus the assumption must have been false, so $p(\mathbf{b}) = 0$.

**Part II: every allocation can be implemented at a core-stable bid profile.** Let $\mathbf{z}$ be an arbitrary allocation at which all the units are allocated.

Define $\mathbf{b}$ such that for all $i \in [n], j \in [K]$, we have $b_{i,j} = M$ if $j \leq z_i$ and $b_{i,j} = 0$ otherwise. Then $\mathbf{x}(\mathbf{b}) = \mathbf{z}$ and $p(\mathbf{b}) = 0$. Let $W = \{i \in [n] \mid z_i > 0\}$ be the set of "winners" at $\mathbf{b}$.

Assume towards a contradiction that $\mathbf{b}$ is not stable. Then there is a blocking coalition $C = \{i_1, \ldots, i_k\} \subseteq [n]$ with alternative bid profile $\mathbf{d} = (\mathbf{d}_{i_1}, \ldots, \mathbf{d}_{i_k})$. Denote $\widetilde{\mathbf{b}} = (\mathbf{d}, \mathbf{b}_{-C})$ the profile where each player $i \in C$ bids $\widetilde{\mathbf{b}}_i = \mathbf{d}_i$ and each player $i \notin C$ bids $\widetilde{\mathbf{b}}_i = \mathbf{b}_i$. We must have $u_i(\widetilde{\mathbf{b}}) \geq u_i(\mathbf{b})$ for all $i \in C$, with strict inequality for some player $i \in C$.

If $C \cap W = \emptyset$, then the only way for at least one of the players in $C$ to change their allocation is to make some of their bids at least $H$. But this increases the price from zero to at least $H$, which yields negative utility for everyone. Thus $C \cap W \neq \emptyset$. We consider two cases:

1. Case $p(\widetilde{\mathbf{b}}) > 0$. Each player $i \in C \cap W$ requires strictly more units at $\widetilde{\mathbf{b}}$ than at $\mathbf{b}$, to compensate for the higher price at $\widetilde{\mathbf{b}}$. Thus $\mathbf{x}_i(\widetilde{\mathbf{b}}) > z_i$ for all $i \in C \cap W$ (†). If $x_i(\widetilde{\mathbf{b}}_i) < z_i$ for some player $i \in W \setminus C$, there would have to exist at least $K + 1$ bids with value at least $H$, and so $p(\widetilde{\mathbf{b}}) \geq H$, which would give negative utility to all the players including the deviators. Thus $x_i(\widetilde{\mathbf{b}}_i) \geq z_i$ for all $i \in W \setminus C$ (‡).

    Combining (†) and (‡) gives a contradiction:

    $$\sum_{i \in C \cup (W \setminus C)} x_i(\widetilde{\mathbf{b}}) > \sum_{i \in C \cup (W \setminus C)} z_i = K.$$

    Thus $p(\widetilde{\mathbf{b}}) > 0$ cannot hold.

2. Case $p(\mathbf{b}) = 0$. Then each player $i \in C \cap W$ requires $u_i(\widetilde{\mathbf{b}}) \geq z_i$. Since $p(\widetilde{\mathbf{b}}) = 0$, the top $K$ bids are strictly positive and the remaining bids are zero. Then each player $i \in C \cap W$ submits exactly $z_i$ strictly positive bids. It follows that $\mathbf{x}_i(\widetilde{\mathbf{b}}) = \mathbf{x}_i(\mathbf{b})$ and $p(\widetilde{\mathbf{b}}) = p(\mathbf{b})$ for each $i \in [n]$, which means no player in $C$ strictly improves. Thus $C$ cannot be blocking.

In both cases 1 and 2 we obtained a contradiction, so $\mathbf{b}$ is core-stable, $p(\mathbf{b}) = 0$, and $\mathbf{x}(\mathbf{b}) = \mathbf{z}$ as required. $\qquad\square$

## D  Theorems from prior work

In this section we include the theorem from [CBL06] that we use. In the problem of prediction under expert advice, there are $N$ experts in total, and at each round $t \in [T]$:

- learner chooses a probability distribution $p_t$ over $[N]$;
- nature reveals the losses $\{\ell_{t,i}\}_{i \in [N]}$ of all experts at time $t$, where $\ell_{t,i} \in [0, L]$.

For a given sequence of probability distributions $(p_1, \cdots, p_T)$, the learner's regret is defined to be

$$\text{Reg}(T, (p_1, \cdots, p_T)) = \sum_{t=1}^{T} \sum_{i=1}^{N} p_t(i) \ell_{t,i} - \min_{i^\star \in [N]} \sum_{t=1}^{T} \ell_{t,i^\star}.$$

For this problem, the exponentially weighted average forecaster, also known as the Hedge algorithm, is defined as follows. The algorithm initializes $p_1 = \sigma$, an arbitrary prior distribution over the experts $[N]$ such that each expert $i$ is selected with probability $\sigma_i > 0$. For $t \geq 2$, the forecaster updates

$$p_t(i) = \frac{p_{t-1}(i) \exp(-\eta \ell_{t-1,i})}{\sum_{j=1}^{N} p_{t-1}(j) \exp(-\eta \ell_{t-1,j})}, \quad \forall i \in [N],$$

where $\eta > 0$ is a learning rate.

Next we include the statement of Theorem 2.2 of [CBL06] in our notation.

**Theorem 7** (Theorem 2.2 of [CBL06])**.** *Consider the exponentially weighted average forecaster with $N$ experts, learning rate $\eta > 0$, time horizon $T$, and rewards in $[0, 1]$. Suppose the initial distribution $\sigma$ on the experts is uniform, that is, $\sigma = (1/N, \ldots, 1/N)$. The regret of the forecaster is*

$$\text{Reg}(T, (p_1, \cdots, p_T)) \leq \frac{\log N}{\eta} + \frac{T\eta}{8}.$$

The following corollary is a well known variant of the above theorem, to allow rewards in an interval $[0, L]$ and an arbitrary initial distribution $\sigma$ over the experts.

**Corollary 1.** *Consider the exponentially weighted average forecaster with $N$ experts, learning rate $\eta > 0$, time horizon $T$, and rewards in $[0, L]$. Suppose the initial distribution on the experts is $\sigma$. The regret of the forecaster is*

$$\text{Reg}(T, (p_1, \cdots, p_T)) \leq \frac{1}{\eta} \max_{i \in [N]} \log\left(\frac{1}{\sigma_i}\right) + \frac{TL^2\eta}{8}.$$

The proof of Corollary 1 is identical to that of Theorem 7, except for the following two differences:

- Instead of $W_t = \sum_{i=1}^{N} \exp(-\eta \sum_{s \leq t} \ell_{s,i})$, we define $W_t = \sum_{i=1}^{N} \sigma_i \exp(-\eta \sum_{s \leq t} \ell_{s,i})$. In this way,

$$\log \frac{W_T}{W_0} = \log W_T \geq -\eta \min_{i^\star \in [N]} \sum_{t=1}^{T} \ell_{t,i^\star} - \max_{i \in [N]} \log \frac{1}{\sigma_i}.$$

- When applying [CBL06, Lemma 2.2], we use the interval for the rewards as $[0, L]$ instead of $[0, 1]$.

