## Roadmap to the appendix

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

} (\frac{2}{3}, \frac{2}{3}, \cdots, \frac{2}{3}, 0, \cdots, 0) & \text{with probability } 0.5 + \delta, \\ (\frac{2}{3}, \frac{2}{3}, \cdots, \frac{2}{3}, \frac{2}{3}, \cdots, \frac{2}{3}) & \text{with probability } 0.5 - \delta, \end{cases}$$

where the number of non-zero entries is $k$ in the first line and $2k$ in the second line, and $\delta \in (0, 1/4)$ is a parameter to be determined later. The randomness used at different times is independent.

- at scenario 2, for every $t \in [T]$, the other bidders' bids are

$$\mathbf{b}_{-i}^t = \begin{cases} (\frac{2}{3}, \frac{2}{3}, \cdots, \frac{2}{3}, 0, \cdots, 0) & \text{with probability } 0.5 - \delta, \\ (\frac{2}{3}, \frac{2}{3}, \cdots, \frac{2}{3}, \frac{2}{3}, \cdots, \frac{2}{3}) & \text{with probability } 0.5 + \delta, \end{cases}$$

where the number of non-zero entries is $k$ in the first line and $2k$ in the second line, and $\delta \in (0, 1/4)$ is a parameter to be determined later. The randomness used at different times is independent.

We denote by $P$ and $Q$ the distributions of $\{\mathbf{b}^t_{-i}\}^T_{t=1}$ under scenarios 1 and 2, respectively, then

$$D_{\mathrm{KL}}(P\|Q) = T \cdot D_{\mathrm{KL}}(\mathrm{Bern}(0.5 + \delta)\|\mathrm{Bern}(0.5 - \delta))$$
$$\leq T \cdot \chi^2(\mathrm{Bern}(0.5 + \delta)\|\mathrm{Bern}(0.5 - \delta))$$
$$= T \cdot \frac{(2\delta)^2}{(0.5 + \delta)(0.5 - \delta)} \leq \frac{64}{3}T\delta^2.$$

Consequently, by [Tsy09, Lemma 2.6],

$$1 - \mathrm{TV}(P, Q) \geq \frac{1}{2}\exp\left(-D_{\mathrm{KL}}(P\|Q)\right) \geq \frac{1}{2}\exp\left(-\frac{64T\delta^2}{3}\right).$$

Next we investigate the separation between these two scenarios. It is clear that

$$\max_{\mathbf{b}_i}\mathbb{E}_P\left[u_i(\mathbf{b}_i; \mathbf{b}^t_{-i})\right] \geq \mathbb{E}_P\left[u_i((1, 1, \cdots, 1, 0, \cdots, 0); \mathbf{b}^t_{-i})\right]$$
$$= \left(\frac{1}{2} + \delta\right) \cdot k + \left(\frac{1}{2} - \delta\right) \cdot \frac{k}{3} = \frac{2 + 2\delta}{3} \cdot k;$$
$$\max_{\mathbf{b}_i}\mathbb{E}_Q\left[u_i(\mathbf{b}_i; \mathbf{b}^t_{-i})\right] \geq \mathbb{E}_Q\left[u_i((1, 1, \cdots, 1, 1, \cdots, 1); \mathbf{b}^t_{-i})\right]$$
$$= \left(\frac{1}{2} - \delta\right) \cdot \frac{2k}{3} + \left(\frac{1}{2} + \delta\right) \cdot \frac{2k}{3} = \frac{2k}{3}.$$

Moreover, under $(P + Q)/2$ (i.e. $\mathbf{b}^t_{-i}$ follows a $\mathrm{Bern}(1/2)$ distribution), suppose that the vector $\mathbf{b}_i$ has $k'$ components smaller than $2/3$. Distinguish into two scenarios:

- if $k' < k$, then

$$\mathbb{E}_{(P+Q)/2}\left[u_i(\mathbf{b}_i; \mathbf{b}^t_{-i})\right] \leq \frac{1}{2} \cdot \frac{2k - k'}{3} + \frac{1}{2} \cdot \frac{2k - k'}{3} \leq \frac{2k}{3};$$

- if $k' \geq k$, then

$$\mathbb{E}_{(P+Q)/2}\left[u_i(\mathbf{b}_i; \mathbf{b}^t_{-i})\right] \leq \frac{1}{2} \cdot (2k - k') + \frac{1}{2} \cdot \frac{2k - k'}{3} \leq \frac{2k}{3}.$$

Therefore, it always holds that

$$\max_{\mathbf{b}_i}\mathbb{E}_{(P+Q)/2}\left[u_i(\mathbf{b}_i; \mathbf{b}^t_{-i})\right] \leq \frac{2k}{3}.$$

Consequently, for each $\mathbf{b}_i$,

$$\max_{\mathbf{b}^\star_i}\mathbb{E}_P\left[u_i(\mathbf{b}^\star_i; \mathbf{b}^t_{-i}) - u_i(\mathbf{b}_i; \mathbf{b}^t_{-i})\right] + \max_{\mathbf{b}^\star_i}\mathbb{E}_Q\left[u_i(\mathbf{b}^\star_i; \mathbf{b}^t_{-i}) - u_i(\mathbf{b}_i; \mathbf{b}^t_{-i})\right]$$