# OpenReview forum: "Learning and Collusion in Multi-unit Auctions"
_NeurIPS.cc/2023/Conference — NeurIPS 2023 poster_

### Official Review · Reviewer_Ge2c · 2023-07-01

**Soundness:** 4 excellent
**Presentation:** 2 fair
**Contribution:** 3 good
**Rating:** 6
**Confidence:** 3

**Summary:**

This paper studies a setting where a single seller runs repeated multi-unit auctions. In each multi-unit auction, there are $K$ identical units of good for sale. Each buyer individually has a valuation for the good with decreasing marginal returns and submits a bid vector. The seller will allocate the units according to the ranking order of the bids, and use uniform pricing. Specifically, the seller will set the price to be the $K$-th highest bid (or $(K+1)$-st highest bid in another variant). Such an auction format takes place in many important real-world settings, such as license auctions for CO2 emissions, ad auctions on online platforms.

The authors first show how to efficiently compute an optimum bid vector in the offline setting, which also serves as a benchmark bidding strategy for performance evaluation in the online setting. In the online setting, they can design algorithms with polynomial running time and low regret under either full information feedback or bandit feedback. They also give a lower bound on the expected regret of this problem. Additionally, they analyze the equilibria in the two variants of the auction to study whether they are susceptible to bidder collusion.

**Strengths:**

1. This work contributes the line of literature on multi-unit auctions by considering dynamic bidding strategies rather than focusing the seller's side. The model is well built on previous work of mechanism design in combinatorial auctions, learning algorithm design in repeated auctions, and has significant implications for real-world issues such as license allocation.

2. The main insight that the bidder can compute an optimum bidding strategy in the offline setting by finding a maximum wight path in a DAG is quite novel. Particularly, the weight of an edge appears to depend on the whole bid vector but turns out to rely only on two neighbor bids, which I find quite interesting. Moreover, it also requires non-trivial techniques to design unbiased estimators for the bandit feedback model.

3. Other characterizations about this problem are provided, including regret lower bounds and equilibrium analysis.

4. All results and proofs are well written.

**Weaknesses:**

The organization of this paper looks weird to me. Section 1.2 covers more than two pages. In particular, "equilibrium analysis" is one of the contributions but all the results about it are piled up in Introduction without an independent section. The paper is titled with ``online learning'', so I assume the algorithms for the online setting should be the highlight of this work, but now Subsubsection 1.2.3 accounts for more than half of Subsection 1.2.

Actually, this part of equilibrium analysis seems to have almost no connection to the $T$-round setting. Theorem 3 and 4 only give regret upper bounds, and do not analyze the convergence of the algorithm. No-regret learning algorithms can converge to a coarse correlated equilibrium, but there is no guarantee of convergence to any pure Nash equilibrium. Therefore, this part in fact studies the pure equilibria in a static setting from the perspective of the seller, aiming to figure out which kind of uniform pricing is a better pricing rule. Maybe the authors should consider changing the title to enlarge the scope of this paper.

**Questions:**

1. I don't see why the authors concludes with a summary that the $K$-th price auction format may be preferable to the $(K+1)$-st price auction. Theorem 6 is a negative result about the core stable profile in the $(K+1)$-st price auction while the $K$-th price auction does not even have a pure Nash equilibrium. Why are they comparable?

2. Minor comments.
- In line 65, $z$ is used to denote an allocation, but then in the following paragraph, $x$ is used to denote the allocation. And $z$ is later used to denote a vertex in DAG.
- I personally feel that "repeated auctions" is the only accurate expression, while "repeated mechanisms" (line 33), "repeated setting" (line 76), "repeated auction" (line 272) are not appropriate expressions.
- Line 195, the angle -> from the angle.
- Line 201, such problem -> such a problem.
- In the definition of $S_i$ (equation 1), no value for $t$ (the superscript) is given.
- The key evaluating metric, regret, is not defined in the model part.
- There are some meaningless repetitions. For example, lines 220-223 almost repeat lines 89-93, the footnote on page 6 repeats the meaning of $x_i$ in line 68.
- Example 1 can help readers to better understand the model, so I think it is better not to defer it to the appendix.
- When $K=1$, the problem then becomes bidding in repeated first-price ($K$-th price) or second-price ($(K+1)$-st price) auctions. Pointing out the special cases may help readers better understand why the $k$-th price auction does not have a pure Nash equilibrium, and why the $k+1$-st price auction is more susceptible to bidder collusion.

---

> ### Author Rebuttal · Authors · 2023-08-09
>
> Thank you for your review.
>
> - Regarding Section 1.2 and Coarse Correlated Equilibria (CCE) and K+1-th pricing rule: Let us first note that in essence, our contribution introduces a learning algorithm for bidding which can enhance comprehension of CCEs within uniform price auctions.  This can be done by simulating our learning algorithm in various settings. This application holds practical significance.
> Moreover, our equilibrium analysis results in Section 1.2 bear substantial value for decision-makers. Specifically, they spotlight a critical distinction: while uniform pricing auctions under the K+1-st pricing rule yield equilibria with zero prices, the same does not apply to the K-th pricing rule. This observation is important given the prevalent use of uniform price auctions.
>
> In any case, we are open to changing the title of the paper to reflect the additional results we have in Section 1.2 for NEs.

---

> > ### Comment · Reviewer_Ge2c · 2023-08-17
> >
> > Thanks for the rebuttal. I don't have any further questions.

---

### Official Review · Reviewer_BfUK · 2023-07-06

**Soundness:** 3 good
**Presentation:** 3 good
**Contribution:** 3 good
**Rating:** 5
**Confidence:** 4

**Summary:**

This paper studies the setting in the multi-unit auction where there are $K$ items to allocate and buyer are not necessarily unit-demand and have quasi-linear valuations with decreasing marginal returns. The bidder set separate bid for each item, and each item goes to its highest bidder, where the price can be either the $K$-th bid, or the $K+1$-th bid. The goal is to design a low-regret algorithm from a bidder's perspective, with the goal of maximizing his utility defined as the value minus payment for the winning items. This paper gives a collection of interesting results:

- a DAG construction of the construction for one bidder, and a proof for the bijection between the bid vectors and paths in the offline setting
- Reduce the learning problem as an online maximum weighted path problem, and a weight-pushing algorithm as a solution to the online full information setting
- Regret lower bounds for the setting.
- Equilibrium analysis of this auction.

**Strengths:**

- This paper provides several interesting results.
- The model is well-motivated and the intro is clean and informative.

**Weaknesses:**

- No empirical experiments, but that's fine since this is a theoretical paper.

Here are some additional comments regarding the presentation:
- The abstract in the pdf version is different from the abstract in the OpenReview system.

**Questions:**

- Is it possible to generalize this regret against an adaptive adversary to the policy regret?
- The equilibrium analysis seems to imply this auction doesn't guarantee any positive revenue, why is it essential to study considered on this result?

**Limitations:**

- no empirical experiments.

---

> ### Author Rebuttal · Authors · 2023-08-09
>
> Thank you for your review.
>
> - Expanding to policy regret: Achieving a deeper understanding of policy regret involves transforming our current setup into a contextual bandit framework with an exponential range of contexts. This challenge presents an exciting avenue for future research.
>
> - Regarding  the Importance of Uniform Price Auctions and Their Variants: It's essential to highlight the distinction between uniform pricing auctions under the K+1-st pricing rule—yielding zero-price equilibria—and the K-th pricing rule, lacking this property. This distinction holds significance when selecting an appropriate uniform pricing variant due to its wide adoption.
> Furthermore, uniform pricing is foundational, achieving equilibrium by matching demand and supply. Its application spans electricity markets, carbon trading, and institutions like the Bank of England and Treasury. The pricing rule in this auction is perceived as fair, and fair pricing mechanisms are particularly relevant in high stakes settings like carbon trading.
> Given these factors, studying uniform price auctions is well-justified. The interplay between pricing variants and their real-world implications underscores the importance of this investigation for informed decision-making.

---

> > ### Comment · Reviewer_BfUK · 2023-08-15
> >
> > Thanks for the rebuttal. I don't have any further questions.

---

### Official Review · Reviewer_5cGR · 2023-07-06

**Soundness:** 3 good
**Presentation:** 2 fair
**Contribution:** 2 fair
**Rating:** 5
**Confidence:** 3

**Summary:**

The paper considers a repeated autcion setting where
the players submit their bids for an item of which $K$ units are avaialble.
An auctioneer computes a price $p$
and allocates the j-th unit to the
owner of the j-th highest bid, charging the price p$ for each unit.
(I did not understand this part completely, see "Questions" below)
The players have quasi-linear valuations with decreasing marginal returns.
The paper considers the problem in both an offline and online settings.
They derive upper boudns on regret for the full information
setting, where the bids are public, and the bandit feedback setting,
where each player only observes the price and their own allocation.


**Strengths:**

The paper considers an interesting problem.
Repeated multi-unit auctions are widely used in practice.
The problem formulation is nice and could lead to interesting follow-up works.


**Weaknesses:**

I found the writing to be somewhat confusing. In particular,
since the our results section does not end till page 4, the model
should have been described in more detail earlier on.


**Questions:**

The disucssion of the model is pretty confusing to me.
How does the auctioneer allocate the units?
The paper repeatedly says
"the j-th unit to the player that submitted the j-th highest bid".
But the players have different bids for different units.

Does the auctioneer simply allocate each item to the player who has the highest
bid for that item given the previous allocations? This would be the most reasonable
thing to do.



**Limitations:**

---

> ### Author Rebuttal · Authors · 2023-08-09
>
> Thank you for your review.
>
> - Regarding the model: The model can be located on page 2 immediately after the introduction. We placed the model early within the paper to facilitate a comprehensive understanding of our contributions within the framework of our proposed model.
>
> - Regarding the allocation rule: The formal depiction of the allocation rule can be found in Section 1.1, while an illustrative example is provided in Example 1 within the appendix. In particular, consider the labeling of units as $1, 2, ..., j, ..., K$. The sentence "the j-th unit to the player that submitted the j-th highest bid" precisely signifies the allocation rule's operation: participants submit their bids, the auctioneer arranges these bids in descending order $(c_1, ..., c_j, ..., c_{n * K})$, and then allocates unit 1 to the bidder with bid $c_1$, unit 2 to the bidder with bid $c_2$, and so on.
>
> - Furthermore, it's important to note that the allocation at time t solely depends on the bids submitted by bidders in round t, excluding any prior rounds. As an example, let's consider K = 2 units and two bidders. In round 5, suppose bidder 1 submits the bid vector [4, 2], and bidder 2 submits [5, 3]. After sorting the bids – [5, 4, 3, 2] – the first unit is allocated to bidder 1, and the second unit goes to bidder 2 (this being the allocation in round 5). Notably, this allocation disregards any events from preceding rounds.

---

> > ### Comment · Reviewer_5cGR · 2023-08-15
> >
> > Thank you for the clarifications; I think the explanation you have written here for the allocation rule is much more clear and it would be good to incorporate it in the paper.
> > I don't have any additional questions.

---

### Official Review · Reviewer_KDVH · 2023-07-06

**Soundness:** 3 good
**Presentation:** 3 good
**Contribution:** 3 good
**Rating:** 7
**Confidence:** 2

**Summary:**

This work systematically considers computational, learning-theoretic, and game-theoretic aspects of multi-unit auctions with uniform pricing. In such auctions, $K$ identical goods are sold to agents with quasilinear utility with a uniform pricing scheme set to either the $K$th or $(K+1)$th highest overall bid (note bidders submit a list of $K$ nonincreasing bids for receiving subsequent goods by diminishing returns). On the computational side, it is shown that the offline optimization problem of maximizing hindsight utility given a history of competing bids (subject to discretization) can be nontrivially and efficiently reduced to computing a max-weight path in a DAG; a similar transformation is then used to devise no-regret algorithms in the online setting in both full-information and bandit information settings. These are complemented by nontrivial regret lower bounds. Finally, this work characterizes the core-stable allocations and prices in these settings to suggest that $K$th price auctions may be more resilient than $(K+1)$th price auctions in practice.

**Strengths:**

The considerations given in this work to this setting are fairly comprehensive on a multitude of axes as listed above. The paper is fairly well-written and provides nice discussion of the problem and related work.

**Weaknesses:**

There remains a fairly large gap between the bandit upper and lower bounds; it's not entirely clear at present whether or not existing learning methods could resolve this gap easily.

**Questions:**

---In general, this paper is quite well-written --- the authors provide nice discussion of the relevance of this setting as opposed to other natural auctions with (implicitly) discriminatory pricing, like GSP, as well as other related work.

---I was not able to verify all the proofs in detail, but the general structure and ideas explained in the text made general sense.

---Are these results robust/naturally extendible in some way to the setting where the parameter $K$ instead is a time-varying parameter $K_t$?

---On the learning-theoretic side, is it clear that existing online shortest path learning methods, i.e. FTPL, cannot work ``out-of-box'' in this setting after applying the (new) reduction? Or does the analysis done here to invoke Hedge guarantees seem necessary to re-do?

---

> ### Author Rebuttal · Authors · 2023-08-09
>
> Thank you for your review.
>
> - About lower and upper bounds: Despite our attempts to enhance these bounds, the ones presented in the paper remain the best results we could attain. Thus we leave this as a future work. Thank you for your understanding.
>
> - Regarding $K_t$: When the number of units $K_t$ is varied over time, our offline algorithm, which leverages a Directed Acyclic Graph (DAG), can be extended by adjusting edge weights according to $K_t$. Furthermore, our learning algorithms, designed for both full information and bandit settings, can be expanded by integrating the modified DAG into an environment where the bidder does not possess knowledge of $K_t$ at the time of bidding in round $t$.
>
> - Regarding FTPL: Yes you are right that FTPL can also achieve a sub-linear regret in $T$ for both full information and bandit settings. However, we found that the regret achieved by FTPL appears to be worse than that achieved by the Hedge algorithm in our problem. Take the full-information setting as an example: by adding independent exponential distributed perturbations with rate $\eta$ to each edge weight, the regret upper bound from the regularizer part scales as $K\log m/\eta$ (m is the number of discretization levels). For the sensitivity part, an upper bound is $K^2 m \eta T$ ($T$ rounds, per-step reward upper bounded by $K$, $Km$ edges in total, and total variation distance $\eta$ between two shifted exponential distributions). Choosing the optimal $\eta$ to balance these two terms gives us a worse regret upper bound than Hedge. Therefore, we decided to use Hedge in this work.

---

> > ### Comment · Reviewer_KDVH · 2023-08-18
> >
> > Thanks for the response (and sorry for the delay in responding)! It may be worthwhile to put a sentence about this last point, that it may be possible to get sublinear regret using off-the-shelf algorithms but would attain a worse rate. But otherwise, I have no further questions and will leave my score as it is.
> >
> > Thanks!

---

> > > ### Author Response · Authors · 2023-08-18
> > >
> > > Thank you for your response! We will incorporate a discussion regarding the final point into the camera-ready version.

---

### Official Review · Reviewer_bLMv · 2023-07-07

**Soundness:** 4 excellent
**Presentation:** 4 excellent
**Contribution:** 3 good
**Rating:** 6
**Confidence:** 3

**Summary:**

This paper studies the problem of learning to bid in a multi-item auction. There are k identical item to be sold. Each bidder has decreasing value v_1, v_2, ..., v_k with value v_j for j'th item allocated to the bidder. The auction allocates to the k highest bidder and charge k'th or (k+1)'th price. It's worth noting that k'th and (k+1)'th price are neither truthful nor first price in this setting. It has the value of being fair - same price is charged to all bidders. However, it is not truthful as the (k+1)'th bid might belong a bidder obtaining some of the items and that bidder might have an incentive to mis-report.
The authors consider two problems in this setting: how does the bidder learn to bid with low-regret in this setting and what type of equilibria can be obtained that are core-stable (no subset of bidders can deviate to obtain a better outcome.
For the first, the authors provide an offline algorithm for computing best-response to a history of bids by the competitors, provide a low regret algorithm when in each round the full vector of bids is revealed and provide a low regret algorithm when in each round only the winning price and the allocation to the individual player is revealed. The authors also provide a lower bound example to bound the minimum regret that is unavoidable.
The authors also study equilibria that are core-stable - however even though this is an auction setting they only consider settings where the players other than the auctioneer deviate. They show that (k+1)'th price can only have zero price pure Nash equilibria that are core stable.

The paper is very well written. All the arguments are detailed and easy to follow. All the proofs are in the appendix but the statements in the main body are clear enough to give the sense of the result and a sketch of how the result is obtained.
For the learning to bid results, the authors map the problem of computing optimal bid to finding a maximum weight path in a directed acyclic graph. With this mapping, they use existing technology to instantiate the hedge algorithm to explore paths. For the bandit setting where only signal about the price and number of units won is available a slower update along the path that won is used.

I am not sure about the value added by the equilibrium analysis. It is not related to the other set of results. Authors show bad properties of the (K+1)'th price auction, and conclude that k'th price auction might be preferred - however that conclusion is not clear.

Update post rebuttal:
Thank you for the rebuttal. It seems fair to restrict to just uniform price auctions. Perhaps the authors can make this more explicit and include further justification in the paper.
For the preference between (K+1)'th, k'th price - I am not sure authors fully addressed this question. Agreed that zero revenue equilibria are bad and the auctioneer would like to know about that, but with the k'th price not even having a pure Nash equilibrium, how should the auctioneer choose? Perhaps the authors should explore non-pure Nash equilibria for k'th price and show that those guarantee non-zero revenue?
It would also be good if the authors could tie together the no-regret analysis with the pure Nash equilibrium analysis - may be by making statements about the coarse-correlated equilibria of the two auction formats.

**Strengths:**

- The paper is well written. Explains all the results well and provides adequate details in the main body and the appendix.
- The algorithms for learning to bid are non-trivial, they build on existing technology but require ideas specific to the model.

**Weaknesses:**

- The results for equilibrium analysis seem unrelated and don't add a lot of value. The conclusion about k'th price auction being preferred is not clear.
- The authors assume that uniform pricing auction is preferred. This could use more justification. Perhaps true first price or truthful payment rules are also worth studying. Would the regret analysis extend to those as well?

**Questions:**

Is the conclusion that k'th price is preferable clear? Is there any other connection to the rest of the paper of this equilibrium analysis?

**Limitations:**

Authors discuss open questions remaining. There is no potential negative societal impact.

---

> ### Author Rebuttal · Authors · 2023-08-09
>
> Thank you for your review.
>
> - Regarding equilibria, the possibility of learning algorithms promoting collusion is a significant worry. Thus, using an auction like the K+1-st, which features equilibria with zero prices, presents issues. These equilibria are not just resistant to coalition deviations but are also quickly identifiable by buyers. While participants might see this as beneficial, it does raise concerns about the auction's effectiveness.
> Delving into the understanding of Nash Equilibria (NEs) further sheds light on worst-case scenarios, providing valuable insights for the auctioneer. An illustrative example of this importance lies in the context of the carbon market. Here, revenue generation stands as a pivotal objective, as the obtained revenue predominantly fuels investments in green technology.
>
> - Regarding K-th price auction being preferred, uniform price auctions utilizing the K-st pricing rule do not allow for an equilibrium with zero revenue. This observation holds significant relevance and should be taken into consideration when deciding upon the appropriate variant of uniform pricing to adopt.
>
> - Regarding uniform pricing: Uniform pricing stands as a fundamental concept, on par with market equilibrium where the demand and supply get matched. Its practical application spans diverse sectors such as electricity, carbon trading, and institutions like the Bank of England and Treasury. Its pricing mechanism is perceived fair, which is important in settings such as carbon trading.
>
> - Regarding regret in auctions beyond uniform pricing: Our analysis covers the special case of K=1, including first price and second price auctions using the K+1st and K-th pricing rules.
> Additionally, our study establishes a strong link to graphical models, suggesting applicability in exploring other auction formats like GSP and GFP. However more analysis is needed to fully solve other formats which we believe would belong to future work.

---

> > ### Comment · Reviewer_bLMv · 2023-08-17
> >
> > Thank you for the clarification.  I do not have any more questions.

---

### Official Review · Reviewer_ZdSw · 2023-07-24

**Soundness:** 3 good
**Presentation:** 3 good
**Contribution:** 3 good
**Rating:** 5
**Confidence:** 2

**Summary:**

In this paper, the author proposes efficient algorithms that players can use for bidding, in both the offline and online settings. Furthermore, the paper shows regret lower bounds and then analyzes the quality of the equilibria in two main variants of the auction. It focuses on studying uniform price auctions the angle of designing bidding algorithms for the players.

**Strengths:**

This paper well points out several limitations regarding multi-unit uniform-price auctions that previous works have not focused on and addresses the novel algorithm for those limitations with online and offline settings for bidding with many theorems and their proofs.

**Weaknesses:**

Regarding a carbon auction, licenses for CO2 emission mentioned in the abstract are not well delivered in the paper. Moreover, The abstract seems to be different from what I see on the review page. Please be consistent with the abstract that the authors deliver.

**Questions:**

Please refer to the weakness section.

**Limitations:**

Please refer to the weakness section.

---

> ### Author Rebuttal · Authors · 2023-08-09
>
> Thank you for your review.
>
> - The uniform price auctions play a pivotal role in determining the allocation of CO2 emission licenses in the EU Emissions Trading System (EU ETS). See also the paper “Reducing Inefficiency in Carbon Auctions with Imperfect Competition, ITCS 2020”, which explains the auction model used for carbon emissions and gives additional references.
> - In these auctions, learning effective bidding strategies is challenging. This is because of the nontruthful nature of the auction and the bid space being exponentially large, which the paper addresses.
> - The relevance of revenue obtained from these auctions holds paramount importance in the context of the carbon market. This revenue can potentially drive investments in green technology and further sustainability initiatives.

---

### Decision · Program_Chairs · 2023-09-21

**Decision:**

Accept (poster)

**Comment:**

Overall, reviewers are all leaning towards acceptance.

Some strengths reviewers mentioned to support the decision:
(1) This paper is well-written.
(2) The problem considered is well-motivated.
(3) The technique to solve the problem is non-trivial and novel.

The reviewers also gave some weaknesses and comments for further improving the paper. In particular, these could be done before publication:
(1) Reviewers have some questions on the connection between the equilibrium analysis and the rest of the paper. Adding some discussions  maybe helpful for readers.
(2) It might be also helpful to add some lines explaining how the conclusion that k-th price is more preferrable after Theorem 6.